# A new model of Notch signalling: Control of Notch receptor cis-inhibition via Notch ligand dimers

Daipeng Chen[1,2‡], Zary Forghany[3‡], Xinxin Liu[3◠], Haijiang Wang[3,4◠], Roeland M. H. Merks[2,5‡]*, David A. Baker[3‡]*

1 School of Mathematics and Statistics, Xi'an Jiaotong University, Xi'an, China, 2 Mathematical Institute, Leiden University, Leiden, The Netherlands, 3 Leiden University Medical Center (LUMC), Department of Cell & Chemical Biology, Leiden, The Netherlands, 4 Department of General Surgery, The First Affiliated Hospital, Xi'an Jiaotong University, Xi'an, China, 5 Institute of Biology Leiden, Leiden University, Leiden, The Netherlands

◠ These authors contributed equally to this work.
‡ RMHM and DAB contributed equally to this work. DC and ZF authors share first authorship on this work.
* merksrmh@math.leidenuniv.nl (RMHM); d.baker@lumc.nl (DAB)

**Data Availability Statement:** All relevant data are within the manuscript and its Supporting information files. Code running on MATLAB

## Abstract

All tissue development and replenishment relies upon the breaking of symmetries leading to the morphological and operational differentiation of progenitor cells into more specialized cells. One of the main engines driving this process is the Notch signal transduction pathway, a ubiquitous signalling system found in the vast majority of metazoan cell types characterized to date. Broadly speaking, Notch receptor activity is governed by a balance between two processes: 1) intercellular Notch transactivation triggered via interactions between receptors and ligands expressed in neighbouring cells; 2) intracellular cis inhibition caused by ligands binding to receptors within the same cell. Additionally, recent reports have also unveiled evidence of cis activation. Whilst context-dependent Notch receptor clustering has been hypothesized, to date, Notch signalling has been assumed to involve an interplay between receptor and ligand monomers. In this study, we demonstrate biochemically, through a mutational analysis of DLL4, both *in vitro* and in tissue culture cells, that Notch ligands can efficiently self-associate. We found that the membrane proximal EGF-like repeat of DLL4 was necessary and sufficient to promote oligomerization/dimerization. Mechanistically, our experimental evidence supports the view that DLL4 ligand dimerization is specifically required for cis-inhibition of Notch receptor activity. To further substantiate these findings, we have adapted and extended existing ordinary differential equation-based models of Notch signalling to take account of the ligand dimerization-dependent cis-inhibition reported here. Our new model faithfully recapitulates our experimental data and improves predictions based upon published data. Collectively, our work favours a model in which net output following Notch receptor/ligand binding results from ligand monomer-driven Notch receptor transactivation (and cis activation) counterposed by ligand dimer-mediated cis-inhibition.

R2021a https://github.com/DaipengChen/A-new-model-of-Notch-signalling.

**Funding:** This work was supported by the Dutch Cancer Society (30861) to DAB, the Nederlandse Organisatie voor Wetenschappelijk Onderzoek grant NWO/ENW-VICI 865.17.004 to RMHM, the Cancer Genomics Centre Netherlands (CGC.NL) to XL. DC and HW were recipients of Chinese Scholarship Council (CSC) funding as part of the CSC Joint PhD Program on Artificial Intelligence and Bioscience between Leiden University and Xi'an Jiaotong University. The funders had no role in study design, data collection and analysis, decision to publish, or preparation of the manuscript.

**Competing interests:** The authors have declared that no competing interests exist.

## Author summary

The growth and maintenance of tissues is a fundamental characteristic of metazoan life, controlled by a highly conserved core of cell signal transduction networks. One such pathway, the Notch signalling system, plays a unique role in these phenomena by orchestrating the generation of the phenotypic and genetic asymmetries which underlie tissue development and remodeling. At the molecular level, it achieves this via two specific types of receptor/ligand interaction: intercellular binding of receptors and ligands expressed in neighbouring cells, which triggers receptor activation (trans activation); intracellular receptor/ligand binding within the same cell which blocks receptor activation (cis inhibition). Together, these counterposed mechanisms determine the strength, the direction and the specificity of Notch signalling output. Whilst, the basic mechanisms of receptor transactivation have been delineated in some detail, the precise nature of cis inhibition has remained enigmatic. Through a combination of experimental approaches and computational modelling, in this study, we present a new model of Notch signalling in which ligand monomers promote Notch receptor transactivation, whereas cis inhibition is induced via ligand dimers. This is the first model to include a concrete molecular distinction, in terms of ligand configuration, between the main branches of Notch signalling. Our model faithfully recapitulates both our presented experimental results as well as the recently published work of others, and provides a novel perspective for understanding Notch-regulated biological processes such as embryo development and angiogenesis.

## Introduction

The ubiquitous Notch pathway is an ancient, highly conserved signalling system whose early appearance in evolution coincided with the emergence of multicellularity [1,2]. It was the first cell receptor signal transduction pathway to be discovered, more than a century ago, and decades of research since then have established that it is a central regulator of cell fate [1,2] that underpins normal embryo development and tissue homeostasis, from controlling the fine-grain patterning of insect eyes and wings, to orchestrating vertebrate segmentation, neurogenesis, angiogenesis, and turnover and differentiation of the gastro-intestinal tract [3–8]. Moreover, corruption of this network has been implicated in numerous pathologies including neurovascular diseases (CADASIL), multisystem disorders (ALAGILLE syndrome) [9] as well as the majority of solid tumours [10–12]. Whilst invertebrates such as *Drosophila* possess a single Notch receptor family member controlled by two cognate ligands, in vertebrates, the Notch pathway is composed of up to four distinct receptor types (Notch1-4) and five different Type 1 transmembrane ligands: Jagged (JAG)1, JAG2, Delta-Like (DLL)1, DLL3, and DLL4 [13,14]. Operationally, the canonical Notch signaling pathway is relatively well characterized. It is activated in a juxtacrine manner through a *trans* interaction between single pass receptors expressed at the surface of one cell and ligands expressed by neighboring cells resulting in structural changes effected by biomechanical strain/pulling forces, which expose specific enzyme cleavage sites [15,16]. Ultimately, a cascade of proteolytic events terminates in the γ-secretase-mediated cleavage of the Notch intracellular domain [17,18], which translocates to the nucleus whereupon it regulates expression of Notch target genes [19,20]. In addition to transactivation, Notch is subject to another major regulatory mechanism termed

cis-inhibition by which ligands block the activity of receptors expressed in the same cell [21,22]. Collectively, these two counterposed processes (transactivation and cis-inhibition) are critical for determining the strength, the duration, the directionality and the specificity of Notch signalling.

In recent years, alongside cell, biochemical and genetic analyses, powerful mathematical approaches coupled to *in silico* modelling have become an important element of the toolkit needed to decipher the molecular details of Notch signalling and to understand the biological consequences of these processes [23–30]. Collier et al. first proposed a mathematical description of lateral inhibition, an evolutionary conserved intercellular signalling mechanism that underlies symmetry breaking in tissues, in which Notch receptor (trans)activation in one cell via ligands expressed by neighbouring cells establishes the differential developmental cell fates necessary for patterning [23]. Whereas this model can recapitulate essential features of transactivation, it was not until the work of Sprinzak et al., which has served as a common starting point for subsequent refinements, that cis inhibition and transactivation were integrated into a single model [22,25]. Latterly, Elowitz and co-workers have developed a new model which takes account of the recently reported phenomenon of cis-activation [31]. Whilst these technical and conceptual advancements are beginning to unravel the deeper complexities of Notch signalling, arguably a significant impediment to obtaining a more complete picture of this vital pathway is the relative paucity of the architectural/molecular details of cis and trans receptor/ligand complexes. Quantitative measurements of Notch/ligand binding have been performed, and structural studies have sought to identify specific binding interfaces [16,32], however, these analyses have relied upon investigating isolated receptor and ligand domains owing to the currently unsurmounted technical difficulties associated with purifying, and structurally and biophysically characterizing full length proteins. One consequence of this, in the absence of available evidence, is that it has been generally assumed that cis and trans receptor/ligand interactions are essentially monomeric. There are, however, sound reasons to suppose that the true picture may be more complicated. Both receptors and ligands harbour multiple EGF-like repeats, which are known to mediate protein-protein interactions [33]. Related to that, we here show biochemically that Notch ligands can efficiently self-associate. This begged the question: what are the potential molecular and biological consequences of Notch ligand oligomerization? Through a combination of experimental approaches and mathematical modelling, we propose a novel view of Notch signalling in which ligand monomer-driven receptor transactivation (and cis-activation) is counterbalanced by ligand dimer-mediated cis inhibition.

## Results

### Notch ligands form dimers/oligomers

To date, it has been assumed that Notch ligands function as monomers. To formally explore this at the biochemical level, we first expressed epitope-tagged ligands in tissue culture cells and tested if the ligands could homo-oligomerize. Fig 1A shows that four different Notch ligands could efficiently self-associate. To further dissect the molecular basis of these interactions, we performed a detailed analysis of the DLL4 ligand. Fig 1B shows, in tissue culture cells, that the DLL4 extracellular domain is necessary and sufficient for homo-oligomerization and that plasma membrane anchorage is not required for this interaction. The DLL4 intracellular domain was found to be dispensable for DLL4-DLL4 binding (Fig 1B). We further demonstrate that whilst cis homo-oligomerization is very efficient (DLL4 molecules expressed in

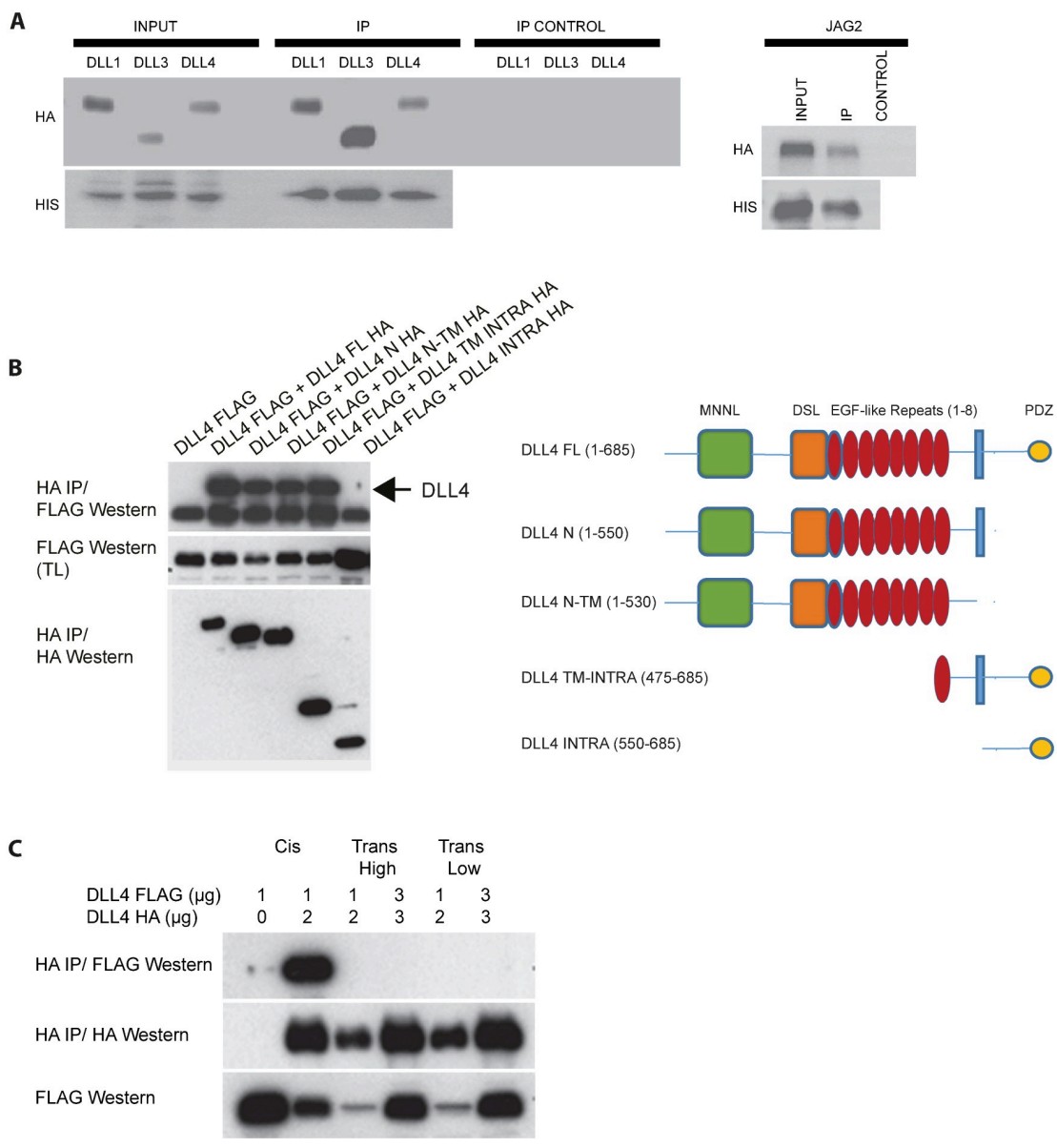

**Fig 1. Notch ligand homo-oligomerization.** (A) HIS epitope-tagged Notch ligands were purified from tissue-culture cells and incubated with the indicated HA-epitope tagged proteins produced by *in vitro* translation. Ligand-ligand interactions were determined by Western blotting using the shown antibodies. (B) Left panel: The indicated constructs were transfected into tissue-culture cells. Complexes were resolved by immunoprecipitation and visualized with the shown antibodies. Right panel: schematic representation of the constructs used in the study. (C) The indicated constructs were transfected into tissue-culture cells in one of three ways: cis- ligands were co-expressed in the same cells sparsely plated to exclude trans interactions; trans (high)- differently tagged ligands were expressed individually in cells, which were subsequently mixed in confluent cell monolayers to enable trans interactions; trans (low)- as for trans (high) but cells were plated at low cell density. Complexes were resolved by immunoprecipitation and visualized using the shown antibodies.

the same cell), trans oligomerization was not observed under the same conditions, though it cannot be ruled out that trans interactions were beyond the detection limit of the experiment (Fig 1C). Collectively, these data show that Notch ligands can forms oligomers both in tissue culture cells and also *in vitro*.

### The membrane proximal DLL4 (EGF-like repeat 8) and the MNNL domain bind to DLL4 *in vitro*

To identify the domains responsible for DLL4 oligomerization, we performed a comprehensive mapping analysis using purified proteins. By these means, we found that EGF-like repeat 8 but not EGF-like repeats 1–7, either as part of the DLL4 extracellular domain (see Fig 2A and 2B) or singularly (Fig 2C), binds to DLL4, consistent with the idea that this domain might underpin DLL4-DLL4 binding (see Fig 2). These experiments also highlighted the MNNL domain as binding efficiently to DLL4 (Fig 2B).

### EGF-like repeat 8 mediates DLL4-DLL4 binding but is dispensable for DLL4 binding to the Notch receptor

To test the requirement of the EGF-like repeat 8 and the MNNL domain for DLL4 oligomerization, we expressed epitope-tagged wild type and mutant DLL4 ligands in tissue culture cells. Whereas the MNNL domain was found to be dispensable for DLL4-DLL4 binding under these conditions, loss of the EGF-like repeat 8 abrogated binding (Fig 3A). Moreover, deletion of EGF-like repeat 7 or EGF-like repeat 6 did not detectably inhibit ligand-ligand binding suggesting that the membrane proximal region encompassing EGF-like repeat 8 encodes a specific DLL4 oligomerization motif (Fig 3B) and that deletion of the EGF-like repeat did not non-specifically corrupt ligand-ligand binding. To determine the impact of deleting the EGF-like repeat 8 on ligand-receptor binding, we co-expressed wild type or mutant DLL4 ligands with Notch 2. Fig 3C shows that DLL4 mutants lacking either EGF-like repeat 8, EGF-like repeat 7, EGF-like repeat 6 or the MNNL domain, associated with Notch2 as efficiently as wild type DLL4. Together, these findings support the view that the DLL4 EGF-like repeat 8 specifically mediates DLL4-DLL4 binding but is not required for Notch receptor-DLL4 binding. Since mutant DLL4 harbouring a deletion of EGF-like repeat 8 presumably exists primarily as a monomer, these results suggest that ligand oligomerization is not a general pre-requisite for Notch receptor binding.

### DLL4 oligomerization is required for cis-inhibition of the Notch receptor

To elucidate the mechanistic consequences of ligand oligomerization, we performed luciferase reporter assays to quantitatively measure Notch receptor activity. When expressed in the same cell as Notch2 receptors, wild type DLL4 or mutant DLL4 ligands lacking either EGF-like repeat 7 or EGF-like repeat 6, all of which can form oligomers, efficiently inhibited the activity of Notch2 in cis. By contrast, DLL4 ligands lacking EGF-like repeat 8, which thus act as monomers, failed to inhibit Notch2 activity when co-expressed in the same cell (Fig 4A). When Notch2 and wild type DLL4 (or mutant DLL4) were expressed in neighbouring cells, we found that Notch receptor transactivation was unaffected by deletions of EGF-like repeat 8, EGF-like repeat 7 or EGF-like repeat 6 (see Fig 4A). Fig 4B shows that deletion of EGF-like repeat 8 did not result in any overt change in the sub-cellular location of DLL4. Overall, these results favour a model in which cis-inhibition, but not transactivation, of Notch signalling specifically depends upon DLL4 ligand oligomerization.

### A general mathematical model describing the potential roles of ligand monomers and dimers in Notch signalling

To further explore the potential biological mechanism of our biochemical findings presented above, we have adapted the mutual inactivation model proposed by Sprinzak et al. [22] to include ligand dimerization and cis-activation. Fig 5 schematically represents possible ligand

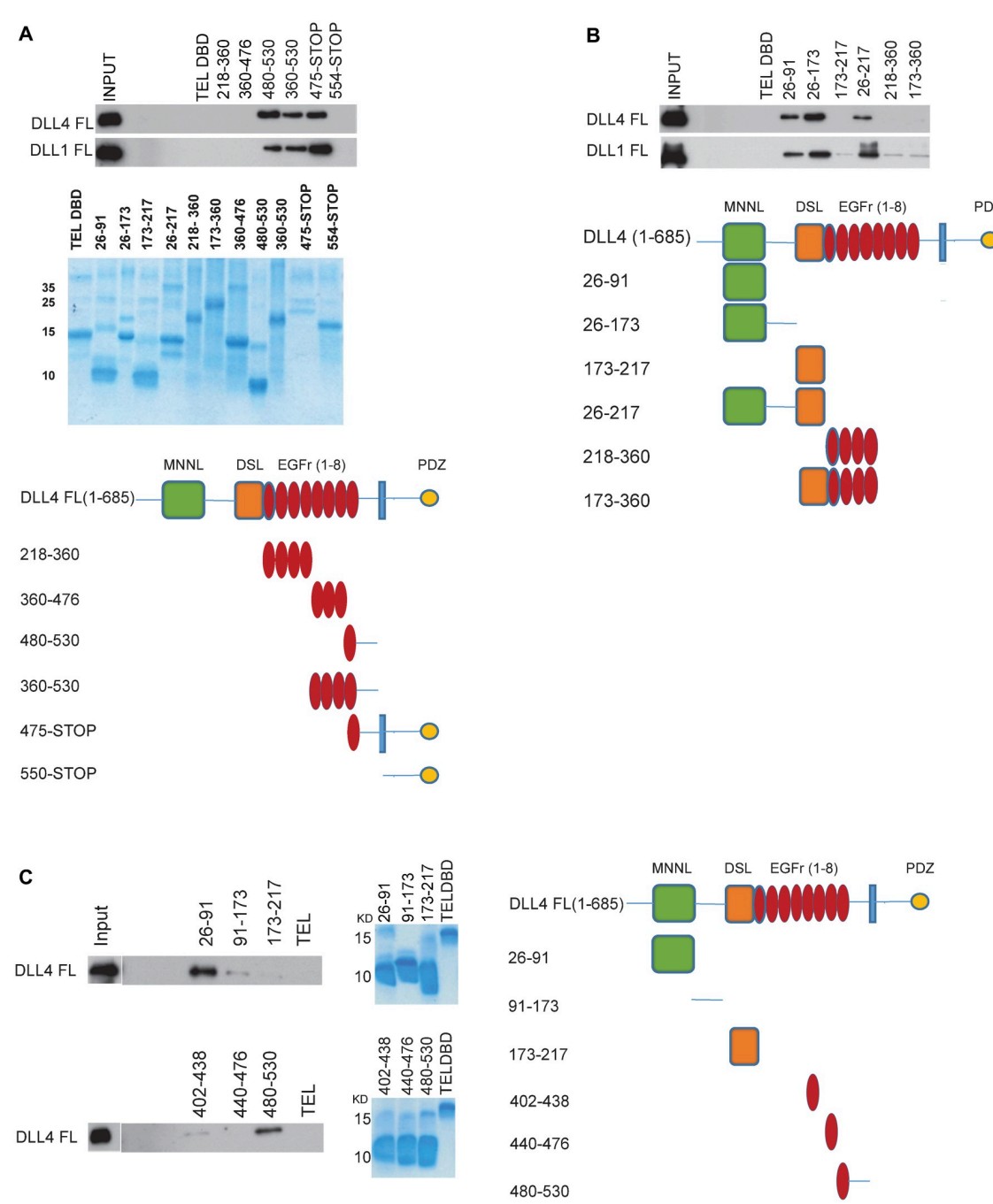

**Fig 2. Biochemical mapping of DLL4 dimerization motifs.** (A-C) The indicated HIS-epitope tagged proteins (a schematic representation of constructs is shown) were purified from *E. coli* (representative Coomassie-stained gels of protein preparations are shown) and incubated with full length DLL4 ligand manufactured by *in vitro* translation. Ligand-ligand Interactions were determined by Western blotting.

and receptor interactions, which underlie these two models, the principal difference being that whereas cis-inhibition is driven via ligand monomers in the mutual inactivation model (Fig 5A), in our general model, cis-inhibition is driven either by ligand monomer or by dimer, or by both (Fig 5B). In any given cell, the new general model is presented mathematically as

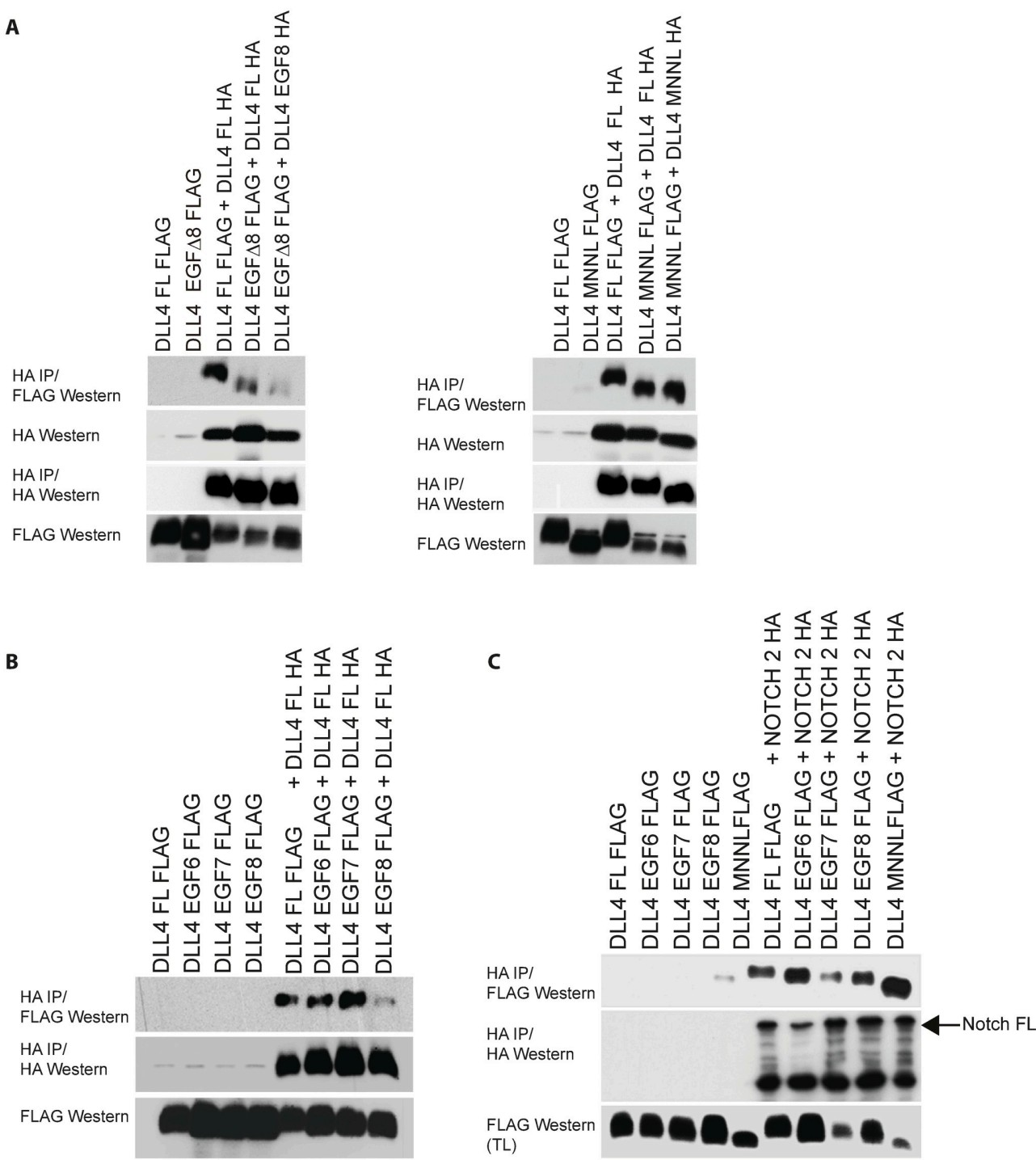

**Fig 3. (A-C) The indicated constructs were transfected into tissue-culture cells.** DLL4 EGF6, EFG7, EGF8, MNNL each have deletions of the named domain. Complexes were resolved by immunoprecipitation and visualized by Western blotting with the highlighted antibodies.

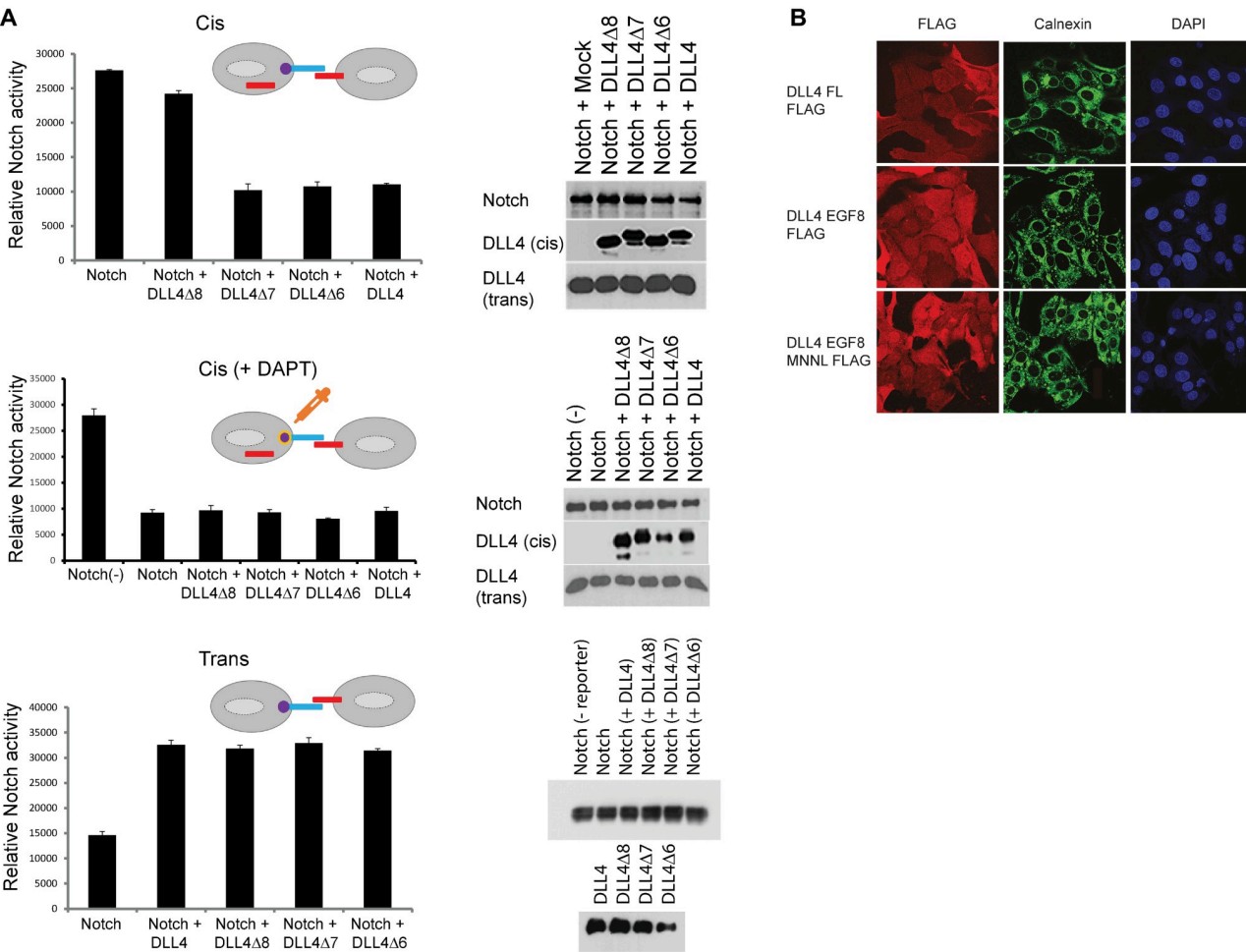

**Fig 4.** (A) Luciferase reporter assays were performed as described in the Methods. Upper graph: U2OS cells co-expressing Notch activity luciferase reporter together with the indicated Notch2 and ligand constructs (cis cells) were co-cultured with cells stably expressing DLL4 (to enable transactivation). Mutant DLL4Δ8 does not cis-inhibit Notch activity significantly (p-value>0.005) whereas other ligands do (p-value<0.005). Middle panel: The same set-up as above, however, cells were cultured in the presence or absence of 10 μM DAPT. Notch (-) means the absence of DAPT. DAPT blocks Notch activity significantly (p-value<0.005) and then cis-ligands do not show inhibition effect further (p-value>0.005). Lower panel: Cells expressing Notch2 and a Notch activity luciferase reporter were separately co-cultured with cells stably expressing the indicated DLL4 constructs. Mutant DLL4Δ8 trans-activates Notch activity significantly (p-value<0.005) and has no significant difference in trans-activation of Notch from other ligands (p-value>0.005). For each analysis, reporter activity was normalized using Renilla luciferase. Levels of ectopically expressed proteins were determined by Western blotting of cell lysates. Each condition in each experiment was performed in triplicate and error bars represent the standard deviation of the mean. Experiments were performed three times. Representative experiments are shown. (B) Immunofluorescence showing the sub-cellular distribution of wild type and mutant DLL4 ligands. The Golgi apparatus was visualized using a calnexin-specific antibody, to rule out aberrant accumulation of ligand.

Eq (1),

$$
\begin{cases}
\dfrac{dL}{dt} = b_L - \beta L - 2k_d L^2 - k_1 N_{ext} L - k_3 NL - k_5 NL, \\[2mm]
\dfrac{dL^*}{dt} = k_d L^2 - \beta L^* - k_2 N_{ext} L^* - k_4 NL^* - k_6 NL^*, \\[2mm]
\dfrac{dN}{dt} = b_N - \beta N - \left(k_1 L_{ext} + k_2 L^*_{ext} + k_3 L + k_4 L^* + k_5 L + k_6 L^*\right)N, \\[2mm]
\dfrac{dS}{dt} = \left(k_1 L_{ext} + k_2 L^*_{ext} + k_5 L + k_6 L^*\right)N - \beta_S S,
\end{cases}
\tag{1}
$$

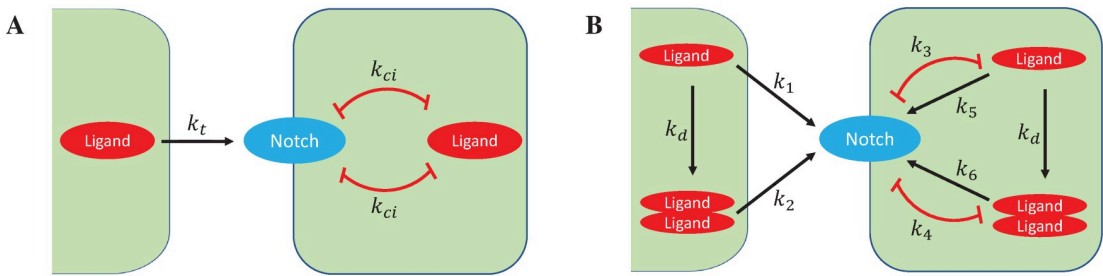

**Fig 5. Diagrammatic representation of the mutual inactivation model and the general ligand dimerization model for Notch signalling.** Black arrows indicate dimerization of ligand monomers, trans-activation and cis-activation of Notch; Red lines represent cis-inhibition.

Where $b_L$ and $b_N$ denote the production rates of Notch ligand ($L$) and Notch receptor ($N$), respectively. The new variable $L^*$, indicating the amount of Notch ligand dimers, and $k_d$, representing the rate of ligand dimerization. The proteins described in the model are assumed to be degraded at a constant rate given by $\beta$ and $\beta_S$, which define the degradation rates of Notch ligands and receptors, and the free Notch intracellular domain ($S$), respectively. In our general model, Notch is trans-activated by ligand monomers (at rate $k_1$) or dimers (at rate $k_2$) expressed in neighboring cells, denoted by $L_{ext}$ (ligand monomer) and $L^*_{ext}$ (ligand dimer), and is cis-inhibited by ligand monomers, $L$, (at rate $k_3$) or dimers, $L^*$, (at rate $k_4$) co-expressed in the same cell. Thus, the general Eq (1), allows for monomer-dependent cis-inhibition, dimer-dependent cis-inhibition or a combination of both. Moreover, ligand monomers or dimers can trigger cis-activation of Notch in the same cell with rates $k_5$ and $k_6$, respectively. The amount of Notch receptors expressed in neighboring cells is denoted by, $N_{ext}$.

In summary, our general model enables us to consider different Notch signalling scenarios. By example, in Eq (1), setting $k_1 = k_t$ and $k_2 = 0$ means that transactivation of Notch is mediated by ligand monomers; $k_3 = 0$ and $k_2 = k_{ci}$ means that cis-inhibition of Notch is mediated specifically and exclusively by ligand dimers; and $k_5 = k_{ca}$ and $k_6 = 0$ means that cis-activation of Notch is mediated by ligand monomers. Alternative scenarios can be tested by varying the value of the corresponding parameters $k_i$(i = 1, 2,.., 6). A comprehensive description of all parameters is detailed in Table 1.

To explore the potential roles of Notch ligand monomers and dimers in Notch signalling, we investigated a number of alternative scenarios in the context of cis-inhibition, trans-activation, and cis-activation. As a first step, we considered the currently accepted overall view of Notch signalling, namely ligand monomer-dependent cis-inhibition in the absence of ligand dimerization ($k_d = 0$) and cis-activation ($k_5 = k_6 = 0$). In this case, the general ligand

**Table 1. Description and baseline values of parameters used in simulations.**

| Parameter | Descriptoon | Values | Units | Source |
|---|---|---|---|---|
| $b_L$ | Baseline production rate of Notch ligands | 200 | molec * hour$^{-1}$ | Estimated from [60] |
| $b_N$ | Baseline production rate of Notch receptors | 200 | molec * hour$^{-1}$ | Estimated from [60] |
| $k_d$ | Baseline oligomerization rate of ligand monomers (n = oligomer size) | $1 * 10^{-4}$ | molec$^{-(n-1)}$ * hour$^{-1}$ | Assumed |
| $k_t$ | Trans-activation rate | $5 * 10^{-5}$ | molec$^{-1}$ * hour$^{-1}$ | [26–28] |
| $k_{ci}$ | Cis-inhibition rate | $6 * 10^{-4}$ | molec$^{-1}$ * hour$^{-1}$ | [26–28] |
| $k_{ca}$ | Cis-activation rate | $5 * 10^{-6}$ | molec$^{-1}$ * hour$^{-1}$ | Estimated from [31] |
| $\beta$ | Degradation rate of typic proteins | 0.1 | hour$^{-1}$ | Estimated from [58] |
| $\beta_S$ | Degradation rate of free Notch Intracellular Domain | 0.5 | hour$^{-1}$ | Estimated from [59] |
| $k_r$ | Transportation rate of proteins from the cytoplasm to the membrane | 0.1 | hour$^{-1}$ | Assumed |

dimerization model (Fig 5B) reduces to a mutual inactivation model (Fig 5A) with $k_1 = k_t$ and $k_3 = k_{ci}$, described mathematically by Eq (2)

$$\begin{cases} \dfrac{dL}{dt} = b_L - \beta L - k_t N_{ext} L - k_{ci} N L, \\[2mm] \dfrac{dN}{dt} = b_N - \beta N - k_t L_{ext} N - k_{ci} L N, \\[2mm] \dfrac{dS}{dt} = k_t N L_{ext} - \beta_S S. \end{cases} \tag{2}$$

Eq (2) can be analytically solved in the steady state $(\bar{L}, \bar{N}, \bar{S})$, leading to the following steady-state levels of Notch ligand and receptor:

$$\bar{L} = \frac{b_L - b_N}{2\beta_1} - \frac{\beta_2}{2k_{ci}} + \sqrt{\left(\frac{b_L - b_N}{2\beta_1} - \frac{\beta_2}{2k_{ci}}\right)^2 + \frac{\beta_2 b_L}{\beta_1 k_{ci}}}, \tag{3}$$

$$\bar{N} = \frac{\beta_1}{\beta_2}\bar{L} - \frac{b_L - b_N}{\beta_2}, \tag{4}$$

where $\beta_1 = \beta + k_t N_{ext}$ and $\beta_2 = \beta + k_t L_{ext}$. When the production rate of ligand is bigger than the Notch receptor production rate ($b_L > b_N$) and the affinity of cis-ligand for Notch receptor is high (e.g., $k_{ci} \gg 4L_0\beta_1\beta_2/(b_L - b_N)^2$), it follows that Eqs (3) and (4) yields the following state:

$$\bar{L} \approx \frac{b_L - b_N}{\beta_1}, \bar{N} \approx 0 \tag{5}$$

Similarly, when the production rate of ligand is smaller than that of Notch receptor ($b_L < b_N$), we find a different state given by:

$$\bar{L} \approx 0, \bar{N} \approx \frac{b_N - b_L}{\beta_2} \tag{6}$$

The mutual inactivation model predicted that Notch receptor and Notch ligand levels are mutually exclusive in the same cell (Eqs 5 and 6; [22]). Consistently, the relative production rates of Notch ligand and receptor determine the output of cell signaling state.

In summary, by exclusively considering ligand monomer-driven cis inhibition of Notch in the absence of ligand dimerization process ($k_d = 0$), our general model essentially reduced to a mutual inactivation model, as defined by Sprinzak et al. [22]. We next tested a number of alternative cases with a well-defined ligand dimerization process ($k_d > 0$) and compared these results to our experimental findings described in Fig 4.

## Exploring the role of ligand monomers and ligand dimers in cis-inhibition and trans-activation

Fig 6A schematically depicts three alternative cases of cis-inhibition. Specifically, in Model C1, ligand dimers mediate cis-inhibition. In Model C2, ligand monomers mediate cis-inhibition of Notch signalling, which is the same as that in the mutual inactivation model. In Model C3, cis-inhibition is mediated by both ligand monomers and ligand dimers. In the context of these models, we do not consider cis-activation ($k_5 = k_6 = 0$) because cis-activation is significantly weaker than trans-activation [31], such that, in this instance, its effects can be ignored. In Fig 6B, we simulated the steady-state levels of Notch activity (free Notch Intracellular domain) driven by the production rate of cis-ligand in the receiving cell (expressing Notch receptor)

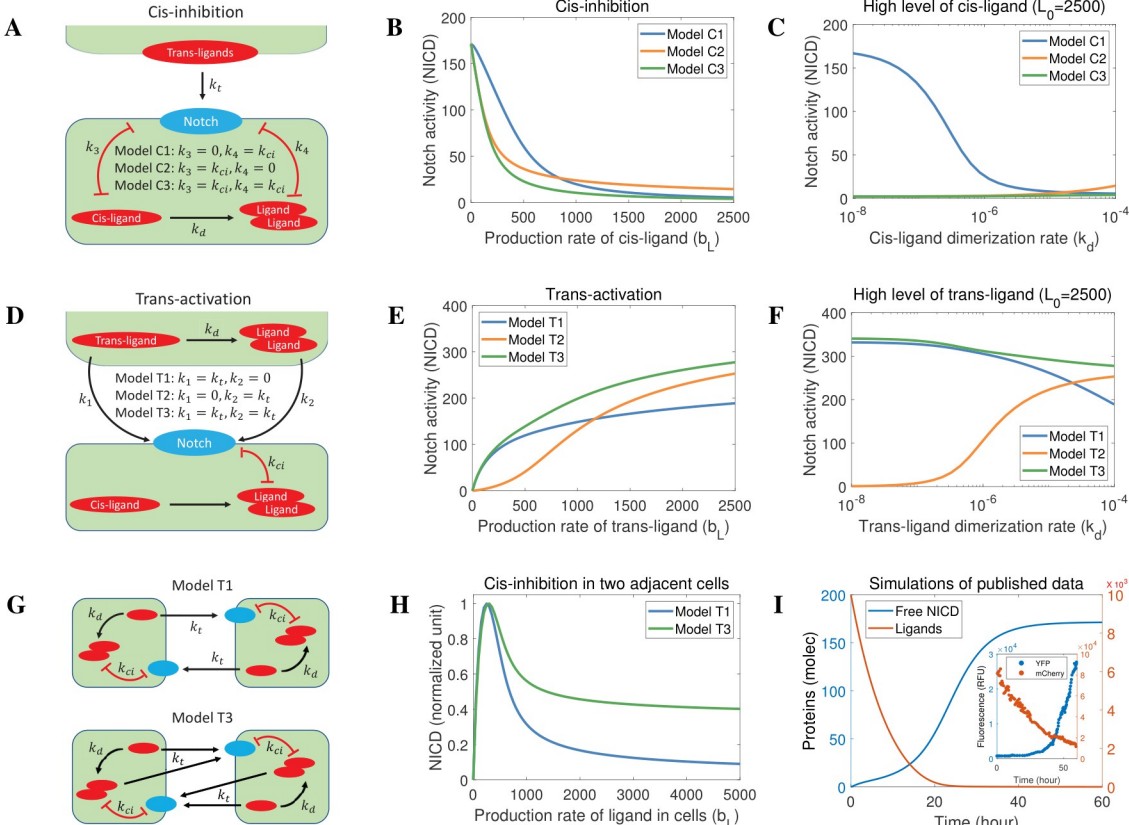

**Fig 6. The role of Notch ligand monomers and dimers in Notch receptor cis-inhibition and trans activation.** (A) The potential models governing cis-inhibition of Notch. The level of trans-ligand in sending cell is fixed whilst cis ligand levels in receiving cell vary as shown. (B) Notch activity in the receiving cell as a function of cis-ligand production rate for the different models shown in A. (C) Notch activity in the receiving cell as a function of cis-ligand dimerization rates for a high level of cis-ligand (the maximum production rate shown in B). (D) The potential models governing trans-activation of Notch. The production rate and dimerization rate of ligand in the receiving cell is fixed whilst trans ligand levels in sending cell vary as shown. (E) Notch activity in the receiving cell in response to increasing trans-ligand production rates for the different models shown in D. (F) Notch activity in the receiving cell in response to decreasing trans-ligand dimerization rates when the level of trans-ligand is high (the maximum production rate shown in E). (G) The potential roles of ligand monomer and ligand dimer governing Notch signaling in two identical cells. (H) Relative Notch activity in response to increasing Notch ligand production rates in two cells for the two cases shown in G. Notch activity is normalized against the maximum Notch activity. (I) *In silico* replication of published cis-inhibition dynamics [22]. In common with their experimental conditions [22], the initial state of ligand levels is high (with a production rate of 0), whilst Notch receptor levels are low.

exposed to fixed levels of trans-ligand in a sending cell. Under these conditions, whether monomer- or dimer-induced cis-inhibition, increase in the production rate of cis ligand resulted in a corresponding inhibition of Notch receptor activity (Fig 6B). Although in model C1 Notch activity decreases (slightly) more slowly with increasing cis-ligand production (cis-inhibition through dimerization) compared to the other models, experimentally it would be hard to distinguish such a small difference. To further investigate the role of ligand monomers and dimers in cis-inhibition of Notch signalling, in Fig 6C, when the production rate of cis-ligand in the receiving cell is sufficient to inhibit Notch receptor activity, we simulated the effects of altering the dimerization rate of cis-ligand in the receiving cell on Notch receptor activity. Clearly, only Model C1, in which ligand dimers but not ligand monomers promote cis-inhibition, faithfully reproduced the experimental observations (Fig 4A), which show that dimer deficient ligands do not efficiently cis-inhibit Notch receptor activation. Therefore, simulations based upon Model C1 (a special case of our general model) match our conclusions

based upon our experimental results, that Notch receptor cis inhibition is ligand dimer- and not ligand monomer-dependent.

Having established the requirement of ligand dimerization for cis inhibition, we next addressed the role of ligand monomers and ligand dimers in trans activation. Fig 6D schematically depicts three alternative cases of trans-activation. In Model T1, ligand monomers mediate trans-activation. In Model T2, ligand dimers mediate trans-activation. In Model T3, both ligand monomers and ligand dimers mediate trans-activation. In Fig 6E, we present the steady-state levels of Notch activity in the receiving cell driven by the production rate of trans-ligand in a sending cell. In common with the simulation described in Fig 6C, in Fig 6F, in the presence of a high production rate of trans-ligand in a sending cell, we measured Notch activity in the receiving cell driven by the dimerization rate of trans-ligand in the sending cell. These results (Fig 6E and 6F) show that both Model T1 and T3 reproduce the Notch receptor trans-activation observed experimentally in Fig 4A, that is, dimer deficient ligands are capable of promoting Notch receptor trans-activation. To further test our model and investigate the potential roles of ligand monomers and dimers in cis-inhibition and trans-activation, we considered two adjacent (identical) cells. Based upon our analyses above, there are two cases (Model T1 and Model T3) governing Notch signalling in two adjacent cells (Fig 6G). In Fig 6H, we present Notch activity driven by increasing production rates of ligands in two cells. We see that Model T1 but not Model T3 shows that higher ligand expression levels elicit greater levels of cis-inhibition in two cells. Collectively, our results provide evidence in favour of a model T1 in which Notch signalling is mediated by ligand monomer-dependent trans-activation and ligand dimer-dependent cis-inhibition.

To further test the general applicability of our model, we ran simulations of our ligand dimerization model (Model T1) to test if it could reproduce previously published experimental data. To measure cis-inhibition and trans-activation experimentally, Sprinzak et al. deployed elegant cell-based reporter assays enabling quantification of both cis-inhibition and trans-activation of Notch signalling [22]. The experimental setting for trans-activation is similar to our diagram depicted schematically in Fig 6A, with fixed levels of trans Notch ligand. Because of the difference in dimensions, it is not possible to parameterize our model using these data. By comparing these simulations to the published data qualitatively, we found that our ligand dimerization model T1 successfully reproduces the reported experimental quantification of both cis-inhibition (Fig 6I) and trans-activation of Notch signalling (Fig A in S1 Text). Thus, in this context, the ligand dimerization model agrees with the mutual inactivation model [22].

Based on our experimental data and numerical simulations, we provide evidence in favour of a model T1 in which Notch signalling is mediated by ligand monomers-dependent trans-activation and ligand dimers-dependent cis-inhibition. Mathematically, this model was derived from the general ligand dimerization model (Fig 5B) by setting $k_1 = k_t$, $k_2 = 0$, $k_3 = 0$, $k_4 = k_{ci}$, $k_5 = 0$ and $k_6 = 0$ in Eq (1), that is

$$
\begin{cases}
\dfrac{dL}{dt} = b_L - \beta L - 2k_d L^2 - k_t N_{ext} L, \\[2mm]
\dfrac{dL^*}{dt} = k_d L^2 - \beta L^* - k_{ci} N L^*, \\[2mm]
\dfrac{dN}{dt} = b_N - \beta N - k_{ci} L^* N - k_t L_{ext} N, \\[2mm]
\dfrac{dS}{dt} = k_t L_{ext} N - \beta_S S.
\end{cases}
\tag{7}
$$

Eq (7) can also be analytically solved in the steady state ($\bar{L}, \bar{L}^*, \bar{N}, \bar{S}$), leading to the following steady-state levels of Notch ligand monomer, ligand dimer and receptor:

$$\bar{L} = \sqrt{\left(\frac{\beta + k_t N_{ext}}{4k_d}\right)^2 + \frac{b_L}{2k_d}} - \frac{\beta + k_t N_{ext}}{4k_d}, \tag{8}$$

$$\bar{L}^* = \frac{k_d \bar{L}^2 - b_N}{2\beta} - \frac{\beta_2}{2k_{ci}} + \sqrt{\left(\frac{k_d \bar{L}^2 - b_N}{2\beta} - \frac{\beta_2}{2k_{ci}}\right)^2 + \frac{\beta_2 k_d \bar{L}^2}{\beta k_{ci}}}, \tag{9}$$

$$\bar{N} = \frac{\beta}{\beta_2} \bar{L}^* - \frac{k_d \bar{L}^2 - b_N}{\beta_2}, \tag{10}$$

where $\beta_2 = \beta + k_t L_{ext}$. Similar to the analysis of Eqs (3) and (4), when the affinity of Notch for ligand dimers is high, Eqs (9) and (10) yields: $\bar{N} \approx 0$ if $b_N < k_d \bar{L}^2$. This indicates that high production rates of Notch ligand could inhibit Notch receptor activity and reduce receptor availability in the same cell, which is consistent with the results (Eq 5) of the mutual inactivation model. However, when the production rate of Notch is large ($b_N > k_d \bar{L}^2$), we have:

$$\bar{N} \approx \frac{b_N - k_d \bar{L}^2}{\beta_2} \tag{11}$$

In contrast to the prediction (Eq 6) of the mutual inactivation model with high Notch expression, here we find that the level of Notch ligand monomer (Eq 8) is independent of the production rate of Notch. In other words, Notch receptor and Notch ligand monomer are not mutually exclusive in the same cell, which can co-express high levels of Notch receptor and ligand simultaneously (Eqs 8 and 11), an observation reported previously [34]. The potential biological implications for the differences in the two models will be considered below.

## Modelling the role of ligand monomers and ligand dimers in cis-activation of Notch signalling

Elowitz and co-workers recently reported cis-activation as a novel, previously overlooked Notch signalling mechanism [31]. In their cis-activation assay conditions, cell-cell contact was eliminated, that is, external ligands and receptors satisfy $L_{ext} = 0$, $L_{ext}^* = 0$ and $N_{ext} = 0$. We have shown that ligand dimers instead of ligand monomers mediate cis-inhibition of Notch, which means $k_3 = 0$ and $k_4 = k_{ci}$ in Eq (1). Consequently, there are three possible scenarios (scenarios 1 to 3 in Fig 7A) for cis-activation of Notch, given mathematically by,

$$\begin{cases} \dfrac{dL}{dt} = b_L - \beta L - 2k_d L^2 - k_5 NL, \\[2mm] \dfrac{dL^*}{dt} = k_d L^2 - \beta L^* - k_{ci} NL^* - k_6 NL^*, \\[2mm] \dfrac{dN}{dt} = b_N - \beta N - k_{ci} L^* N - k_5 LN - k_6 L^* N, \\[2mm] \dfrac{dS}{dt} = k_5 LN + k_6 L^* N - \beta_S S, \end{cases} \tag{12}$$

with $k_5 = k_{ca}$ and $k_6 = 0$ for scenario 1; $k_5 = 0$ and $k_6 = k_{ca}$ for scenario 2; $k_5 = k_{ca}$ and $k_6 = k_{ca}$ for scenario 3. By running simulations using reference parameter values (Table 1), in Fig 7B–7E we demonstrated that the experimental data (non-monotonic response of Notch activity to

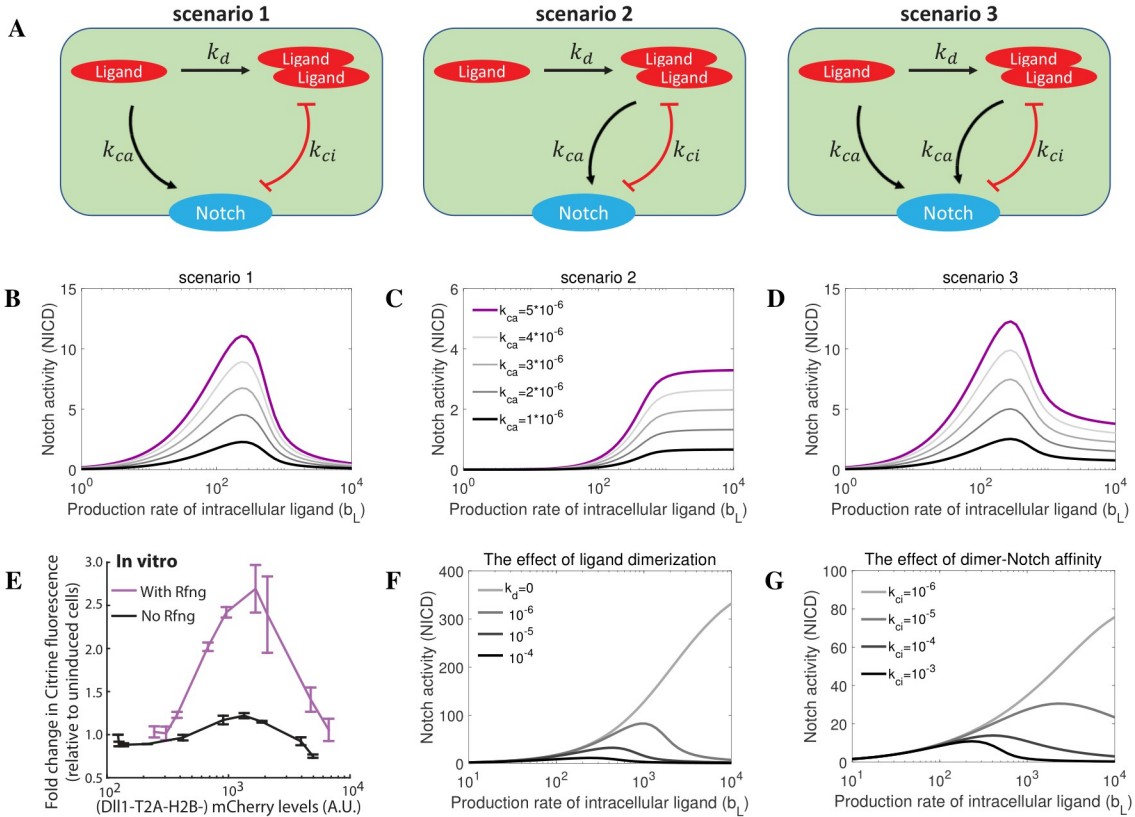

**Fig 7. The role of Notch ligand monomers and dimers in cis-activation.** (A) Schematic representation of different potential rules governing receptor/ligand interactions in Notch signaling. Scenario 1: monomer mediates cis-activation and dimer mediates cis-inhibition; scenario 2: dimer mediates cis-activation and cis-inhibition; scenario 3: monomer mediates cis-activation, dimer mediates both cis-activation and cis-inhibition. (B-D) Simulations of cis-activation for each of the scenarios. Different cis-activation rates are tested. (E) Published in vitro cis-activation experiments [31]. The response of Notch to cis ligand level is non-monotonic. (F) The role of Notch ligand dimerization in cis-activation. Notch ligand dimer does not directly mediate cis-activation, but ligand dimerization is necessary to explain the experimental observations. (G) Lower affinity of Notch for ligand dimer promotes cis-activation and limits cis-inhibition of Notch signalling.

cis-ligand levels in Fig 7E) is best explained by the ligand dimerization model with the parameters setting in scenario 1, which assumes that ligand dimers mediate cis-inhibition whilst ligand monomers mediate cis-activation. In scenario 3, Notch signalling cannot be limited to a low level by high production rate of cis ligand because of ligand dimers-mediated cis-activation (Fig 7D). To explain the observed non-monotonic response of Notch receptor to cis ligand levels (Fig 7E), Nandagopal et al. [31] proposed a number of possible mechanisms, but did not include the explicit ligand dimerization step that our data suggest. Interestingly, we predict that the rate of ligand dimerization modulates both the width and the amplitude of the Notch cis-activation peak, as well as the ligand concentration at which maximum cis-activation is reached (Fig 7F). Additional numerical simulations predict that a Notch receptor could show stronger cis-activation and weaker cis-inhibition if the affinity of Notch for ligand dimers is lower (Fig 7G), which is consistent with the reported experimental results [31]. In summary, our simulations suggest that cis-activation is mediated by ligand monomers, but the non-monotonic response of Notch to cis ligand levels is dependent on ligand dimers-mediated cis-inhibition.

## Modelling predictions are independent of ligand oligomer size and subcellular localization for ligand dimer-mediated cis-inhibition

It is currently not possible biochemically to quantitatively distinguish between ligand homo-oligomerization and homo-dimerization. To test if our modelling prediction would be affected by ligand oligomer size ($n$, the number of monomers composing an oligomer), we extended the ligand dimerization model T1 to include ligand oligomerization (Fig B in S1 Text), given mathematically by,

$$\begin{cases} \dfrac{dL}{dt} = b_L - \beta L - P(L, n) - k_t N_{ext} L - k_{ca} N L, \\[2mm] \dfrac{dL^*}{dt} = Q(L, n) - \beta L^* - k_{ci} N L^*, \\[2mm] \dfrac{dN}{dt} = b_N - \beta N - k_{ci} L^* N - k_t L_{ext} N - k_{ca} L N, \\[2mm] \dfrac{dS}{dt} = k_t L_{ext} N + k_{ca} L N - \beta_S S, \end{cases} \tag{13}$$

where $P(L, n) = n k_d L^n + (n-1) k_d L^{n-1} + \cdots + 2 k_d L^2$ and $Q(L, n) = k_d L^n + k_d L^{n-1} + \cdots + k_d L^2$ represent the process of Notch ligand oligomerization. Here $L^*$ is the quantity of ligand oligomers, of all possible sizes, in cells. To exclude the mixing effect of oligomers of different size, we only considered the homogeneous populations of oligomer of defined size in each simulation, which means $P(L, n) = n k_d L^n$ and $Q(L, n) = k_d L^n$ in numerical simulations. In Fig 8A–8C, we present Notch activity (normalized) driven by the production rate of intracellular ligands in different experimental settings. These results clearly demonstrate that the Notch receptor activity is independent of oligomer size (dimer, trimer or tetramer) qualitatively. Thus, the

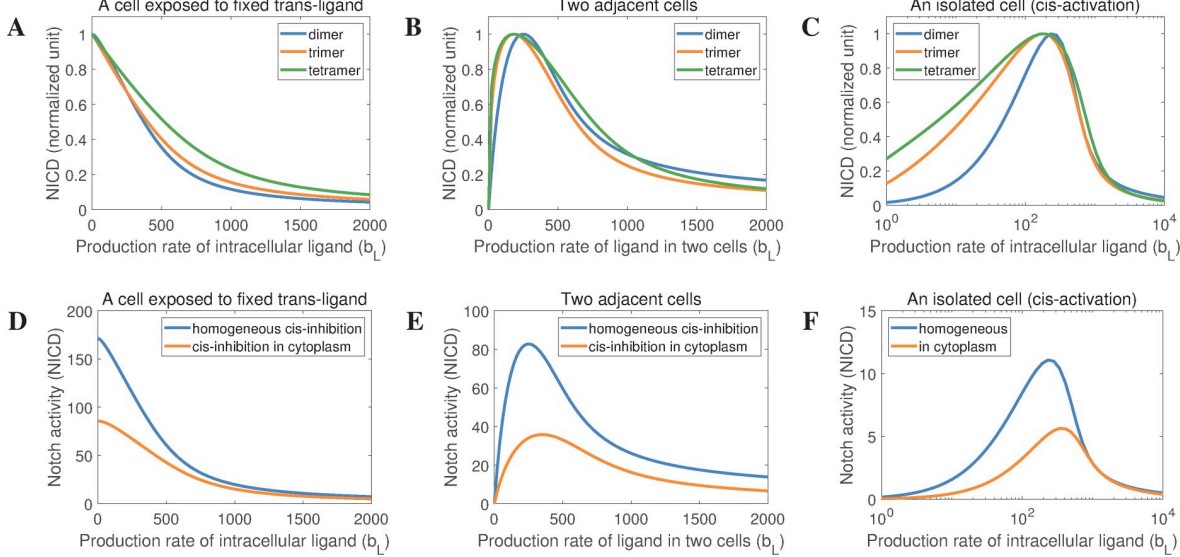

**Fig 8. The effect of ligand oligomer size and subcellular localization on cis inhibition of Notch signalling.** (A-C) The response of Notch to intracellular ligands of different oligomer size in a cell exposed to fixed levels of trans-ligand (A), in two identical cells (B), and in a single cell (C). Notch activity is normalized against the maximum Notch activity. (D-F) The response of Notch to the production rate of intracellular ligand. Here, we compared simulations of cis inhibition which can occur in all cell compartments (spatially homogeneous), with cis-inhibition occurring only in the cytoplasm.

predictions of the Notch ligand dimerization model are general and representative across a range of oligomer sizes.

Another interesting question relates to the subcellular compartment in which cis-inhibition occurs. To investigate the effect of the subcellular localization of ligand dimerization and dimer-mediated cis-inhibition on Notch signalling, we have developed a mathematical model which distinguishes the proteins found in the cytoplasm (with subscript $c$) and the proteins expressed on the cell membrane (with subscript $m$). Based on previous studies [31, 35], this model makes the following initial assumptions: ligand dimerization and ligand dimer-mediated cis-inhibition of Notch occurs in the cytoplasm; trans-activation and cis-activation happens at the membrane. This is expressed mathematically by,

$$
\begin{cases}
\dfrac{dL_c}{dt} = b_L - \beta L_c - 2k_d L_c{}^2 - k_r L_c, \\[2mm]
\dfrac{dL^*}{dt} = k_d L_c{}^2 - \beta L^* - k_{ci} N_c L^*, \\[2mm]
\dfrac{dN_c}{dt} = b_N - \beta N_c - k_{ci} L^* N_c - k_r N_c, \\[2mm]
\dfrac{dL_m}{dt} = k_r L_c - \beta L_m - k_t N_{ext} L_m - k_{ca} N_m L_m, \\[2mm]
\dfrac{dN_m}{dt} = k_r N_c - \beta N_m - k_t L_{ext} N_m - k_{ca} L_m N_m, \\[2mm]
\dfrac{dS}{dt} = k_t L_{ext} N_m + k_{ca} L_m N_m - \beta_S S.
\end{cases}
\tag{14}
$$

In this model, Notch ligands and receptors are produced in the cytoplasm, and non-cis-inhibited ligands/receptors are subsequently transported from the cytoplasm to the membrane (see Fig B in S1 Text) with a transportation rate $k_r$. Other parameters are the same as those mentioned in Table 1. Based on Eq (14), we simulated Notch activity driven by the production rate of cytoplasmic ligands in different experimental settings. In Fig 8D–8F, we compared the response of Notch to increasing production rates of cytoplasmic ligands for the model where ligand dimerization and dimer-mediated cis-inhibition of Notch occurs in the cytoplasm (Eq 14) and the model where cis inhibition is spatially homogeneous (Eq 13). Although cis-inhibition in the cytoplasm appears to inhibit Notch signalling more readily (Fig 8D–8F), this difference is parameter-dependent. Qualitatively, these simulations demonstrate that the modelling predictions do not depend on the subcellular localization of ligand dimerization and dimer-mediated cis-inhibition of Notch.

## Modelling the role of dimer-dependent cis-inhibition in tissue patterning

The preceding analyses considered signalling mechanisms controlled by ligand dimer formation at the molecular and individual cell level. We next tested the validity of our model in the context of characterized tissue patterning processes. Whilst, it is established that the Notch pathway is indispensable for tissue development and maintenance in all known metazoans studied to date, historically, the fruit fly has proven to be an invaluable tool for delineating the basic components and modus operandi of the Notch network generally, and cis inhibition in particular. Indeed, the name Notch derives from the characteristic pattern of wing 'Notches' exhibited by the wings of Drosophila encoding Notch mutations, first recorded a century ago.

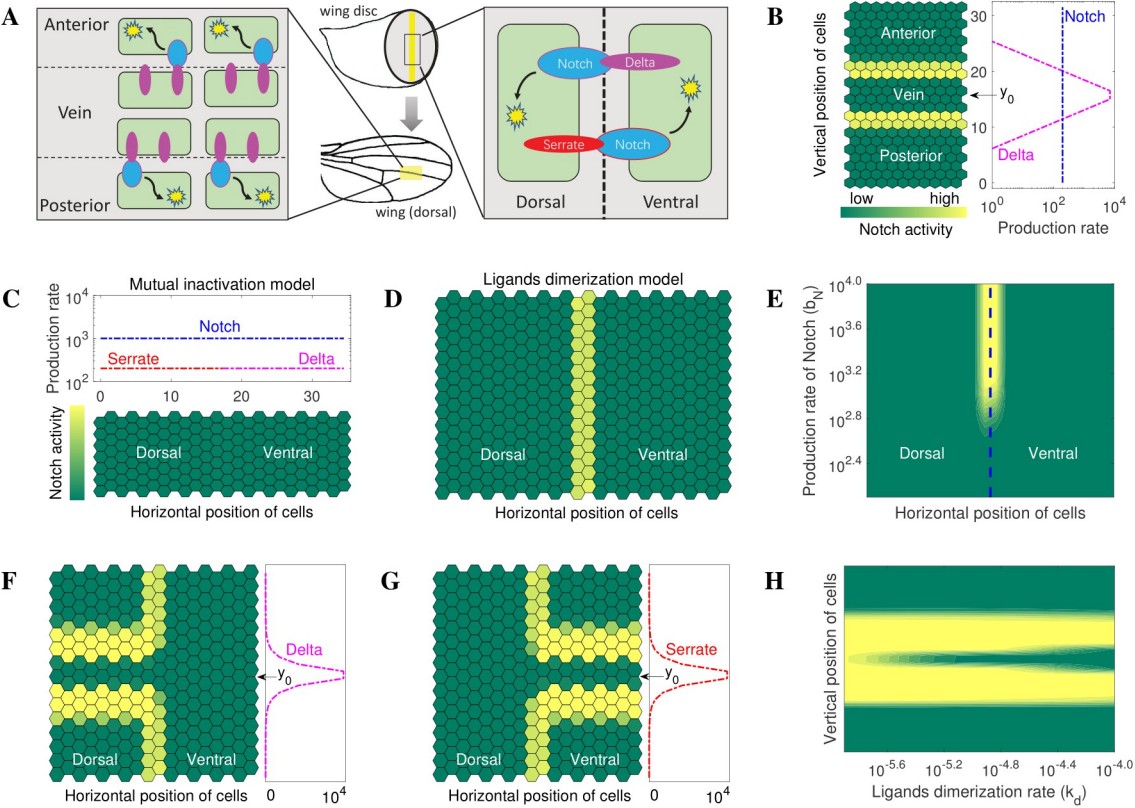

**Fig 9. Modelling multicellular patterning.** (A) Schematic representation of Notch signalling along the wing vein and along the dorsoventral boundary of the wing disc. Blue ovals represent Notch. Red ovals represent serrate. Purple ovals represent Delta. Notch activity is shown by yellow stars. (B) *In silico* replication of Notch signalling along the wing vein (for movie of the simulation, see S1 Video). The production rates of Delta decay exponentially from the center $y_0$, according to $b_L(y) = L_{max}/\exp(|y - y_0|)$. (C-H) *In silico* replication of Notch signalling around the dorsoventral boundary (for movie of the simulation, see S2 Video). (C-D) The ligand dimerization model, but not the mutual inactivation model, replicates the observed *in vivo* pattern of Notch signalling along the dorsal-ventral boundary in Drosophila wing discs. (E) Notch signalling at the dorsal-ventral boundary for different production rates of Notch show loss of the dorsal-ventral boundary as a function of decreasing Notch production rate, consistent with the results of *in vivo* experiments [45, 46]. (F-G) Ectopic expression of ligands induces Notch activity in dorsal cells (mediated by Delta) or ventral cells (mediated by Serrate) adjacent to the ectopic expression stripe where the production rates of ectopic ligands decay exponentially from the center $y_0$, similar to the profile described in B. (H) High production rates of Notch ligands stimulates ectopic Notch receptor activity when the ligands dimerization rate is low, consistent with previously published experimental data [43].

Notch signalling underpins two patterning processes occurring in two perpendicular planes during wing morphogenesis: anterior/posterior signalling required for the formation of wing veins (Fig 9A); and dorsal-ventral patterning needed to establish the wing margin (Fig 9A). To model the role of ligand dimers in wing vein patterning, we considered a gradient of Notch ligand, and a constant level of Notch receptor, across multiple cells yielding a central region of peak ligand concentrations, which delimited the site of vein formation through activation of Notch signalling in neighboring cells (expressing relatively lower levels of ligand, see Fig 9B). Our simulations show that the ligand gradient promoted two parallel bands of Notch signalling, which border, and thereby define the position of vein formation (Fig 9B). These results recapitulate observed *in vivo* patterning processes [36], which have also been successfully reproduced by other mathematical models, notably the mutual inactivation model [22,25].

In contrast to wing vein formation, dorsal-ventral patterning of the Drosophila wing, in principle, presented a challenge to the mutual inactivation model since this model essentially precludes the co- availability of high levels of ligand monomers and Notch receptors in the

same cells (see Eqs 5 and 6), an experimentally demonstrated prerequisite for this patterning [34]. As mentioned previously, no such constraints exist in the dimerization model T1 (see Eqs 8 and 11). During the early stages of wing development, a dorsal-ventral boundary (later the wing margin) divides wing cells into a dorsal compartment and a ventral compartment. Notch appears to be uniformly expressed in all cells whereas the Notch ligands are differentially expressed: Serrate expression is restricted to mainly dorsal cells, whilst functional Delta is localized to ventral cells (although Delta is also expressed in dorsal cells) [37–39]. To test if our model could reproduce wing dorsal-ventral patterning, we performed simulations in which Serrate and Delta were expressed in the dorsal and ventral compartments, respectively, whereas Notch was uniformly expressed throughout both compartments (Fig 9C). In basic simulations (Fig 9C and 9D), we assume that Notch levels are 'in excess' of ligand levels to enable a 'competent' receiving cell signaling state (Eqs 6 and 11). This assumption is required by both the mutual inactivation model (Eq 6) and our model (Eq 11). It has been established experimentally that the glycosyltransferase, Fringe, which is expressed dorsally, mediates the glycosylation of Notch and thus increases the affinity of Notch for Delta but reduces it for Serrate [40,41]. Therefore, in our simulations, Notch in dorsal cells can only be trans-activated by Delta whereas Notch in ventral cells can only be trans-activated by Serrate, which is consistent with the experimental results induced by ectopic expression of Delta or Serrate in dorsal and ventral compartments [34,37,42,43].

Fig 9C, shows that a mutual inactivation model cannot generate a Notch-signalling-dependent dorsal-ventral boundary necessary for the establishment of a wing margin. As alluded to earlier, this is because Notch receptors and ligands are mutually exclusive within the same cell (Eqs 5 and 6). Consequently, there are insufficient levels of ligands in cells to trans-activate Notch signalling across the dorsal-ventral border. Lebon et al. [44] proposed a model, including Fringe-dependent cis-inhibition, to explain dorsal-ventral boundary patterning. Although reduced cis-inhibition in glycosylated dorsal cells could allow high levels of Serrate and Notch to coexist, Delta and Notch in ventral cells are still mutually exclusive. Therefore, it is difficult to generate stable, bidirectional Notch signalling at the dorsal-ventral interface. Strikingly, this is not the case for the ligand dimerization model T1. Instead, we observe the robust dorsal-ventral patterning characteristic of normal wing morphogenesis observed *in vivo* (Fig 9D; [34,37,42,43]). Several additional simulations further highlight the power of the ligand dimerization model (Fig 9E–9H). Decreasing the production rate of Notch results in the disappearance of Notch signaling at the dorsal-ventral border (Fig 9E), consistent with the classic Notch phenotype at the wing margin induced by a loss-of-function Notch mutation *in vivo* [45,46]. In Fig 9F and 9G, we replicated the experimental pattern of Notch signalling induced by ectopically expressed Delta or Serrate [34,37,42,43]. Finally, our model simulations showed that high production rates of ligands, when the ligand dimerization rate is low, stimulated ectopic Notch receptor activity (Fig 9H). These results closely match the experimental findings of Flemming et al. [43]. Interestingly, they showed, by means of expressing, during Drosophila wing disc development, serrate ligand mutants harbouring specific EGF domain deletions, that trans activation and cis inhibition could be uncoupled, that is, certain mutants could selectively promote Notch receptor trans activation but not execute cis inhibition.

Collectively, these results suggest that the ligand dimerization model can recapitulate *in vivo* tissue patterning processes.

## Discussion

In this study, we have taken a combined experimental and mathematical modelling approach to establish biochemically that Notch ligands can self-associate, and to dissect the potential

role of this phenomenon in Notch signal transduction. Several lines of evidence are presented, centred chiefly on the Notch ligand, DLL4, which support the view that net signalling output from the Notch pathway is the product of ligand monomer-mediated trans activation (and cis-activation) counterposed by ligand oligomer-mediated cis inhibition of Notch receptor activity. Our new model is the first to propose a concrete molecular distinction, in terms of ligand configuration, between the two principal branches of Notch signalling, specifically that whereas ligand monomers promote Notch receptor transactivation, they are insufficient to account for cis inhibition which is induced via ligand dimers. Our model exhibits two features that are consistent with our molecular findings (see Figs 1–3). Firstly, cis inhibition is driven by ligand dimers, which can associate independently of receptor binding prior to receptor/ligand complex formation, that is, cis inhibition is not (strictly) a stepwise process of monomer to dimer transitions assembled on preformed receptor/ligand complexes at the cell membrane. Secondly, our modelling results do not depend on cis inhibition occurring at a particular sub-cellular compartment, such as the cell membrane. Rather, cis inhibition can also occur cytoplasmically (Fig 8D–8F), in line with our observations that ligands lacking a transmembrane domain can efficiently self-associate and execute Notch receptor cis inhibition (see Figs 1 and 4). These data are in agreement with earlier studies demonstrating cytoplasmic receptor/ligand binding [35] and raise the possibility that cytoplasmic ligand dimer-dependent cis inhibition of Notch activity could serve to block 'mis-firing' of the receptor prior to its expression at the plasma membrane. Related to this, whilst it is clear that ligand dimers are necessary to drive cis inhibition, it is not clear whether both ligand dimers and monomers are capable of stimulating receptor transactivation. To address this mathematically, we considered two cases: dimers promote cis inhibition/monomers promote transactivation; dimers promote cis inhibition/monomers and dimers can promote transactivation (Fig 6G). Our numerical simulations favour the former case (see Fig 6H). In further support of this idea is our finding that ligands harbouring mutations that abrogate self-association, stimulated Notch transactivation as efficiently as wild type ligands (see Fig 4A). The general applicability of our model at both the cellular as well as the tissue level was demonstrated by the fact that it can recapitulate the results of previously published work (Figs 6I, 7B and 9, and A in S1 Text).

Nandagopal et al. recently unveiled a previously overlooked dimension of Notch signalling, termed cis activation, which results from monomeric interactions between receptors and ligands expressed at the plasma membrane of the same cell [31]. Our model could faithfully recapitulate the published experimental data (see Fig 7). Intriguingly, their mathematical model postulates that cis activation becomes cis inhibition, in a ligand concentration-dependent fashion, following binding of additional ligand monomers to ligand monomers complexed with Notch receptors at the cell membrane (presumably prior to receptor-bound monomer promoting cis activation). In light of this, and given the results of our own simulations, our model suggests that ligand dimerization may ensure a correct balance between cis inhibition and cis activation/trans activation (see Fig 7F).

To date, a major gap in Notch signalling knowledge is the precise nature of ligand, receptor and receptor/ligand complexes at a detailed molecular/structural level encompassing protein-protein interactions and the precise role of post-translation modifications such as glycosylation [47]. In particular, it is unknown if trans receptor/ligand interactions differ conformationally from cis receptor/ligand interactions, and whether or not this could underlie or at least contribute to their distinct effects on receptor activity. Whilst such architectural details are currently lacking, a notable result of our mathematical modelling is that the size of ligand oligomers does not change the predictions of Notch receptor activity based on the ligand dimerization model (see Fig 8A–8C). Additionally, Notch ligand oligomerization could have major implications for our understanding of the dynamics of the Notch pathway because

regulation of dimer formation/dimer disassembly, might represent an extra point of Notch signalling strength control.

There are four mammalian Notch receptors and five mammalian Notch ligands, and the manifold potential receptor-ligand combinations will give rise to the different signalling outputs necessary for tissue patterning. It is established that different ligands can elicit unique cell fates. By example, JAG1 and DLL4 exert opposing effects on angiogenesis [48]. Moreover, recent work has established that ligands can stimulate either discrete pulses of Notch activity (in the case of DLL1) or a sustained period of signalling (in the case of DLL4) yielding distinct gene expression outcomes [24]. In this context, two other potential facets of Notch signalling merit consideration. Whilst here we have uncovered a mechanistic role for ligand oligomerization, there is growing evidence that Notch receptors can also oligomerize/dimerize though the precise molecular consequences of this remains elusive [49,50]. Moreover, since all Notch ligands share a common overall architecture, it could be of interest to investigate if other Notch ligands behave similarly to DLL4 at the biochemical level and, furthermore, if there is a biological role for ligand hetero-oligomerization. Interestingly, simple biochemical experiments revealed that ligands can indeed hetero-oligomerize (see Fig C in S1 Text). Related to this, experiments testing the role of these phenomena in the activity of other Notch receptors could further illuminate the process. Such investigations coupled to our refined mathematical models will help to fully disentangle this core signalling pathway.

By adding Jagged dynamics to the mutual inactivation model, Boareto et al. predicted new hybrid sender/receiver (tip/stalk) cellular phenotypes [27] and proposed that this hybrid cellular state plays an important role in tumour angiogenesis [28]. In the context of Epithelial-to-Mesenchymal Transition (EMT), this hybrid cellular phenotype explains the presence of tumour transition states during EMT [51], which have been experimentally identified [52]. Thus, it would be an interesting to consider Jagged dynamics in our ligand dimerization model and to test if this coupling is able to incorporate hybrid cellular phenotypes.

In summary, we have delineated a previously unreported requirement for ligand dimerization in the cis inhibition of Notch receptor activity. This new mechanism could help determine the strength, the direction, the specificity and the nature of the output of the Notch signalling pathway. A novel mathematical model has been developed which successfully captures this process at the molecular and cellular level and, compellingly, can reproduce *in vivo* tissue patterning processes [34,37,42,43]. One important feature of our model is that even when Notch levels are high the ligand monomer is free to activate neighbouring cells (Eqs 8 and 11). This differs from previous models and such implications can be studied experimentally, as shown in Fig 9. It will be of interest to test if the same model can provide important insights into Notch-controlled physiological processes such as sprouting angiogenesis [53–55], and, given the global interest in the role of Notch signalling in human pathogenesis [56,57], to determine how these ideas will impact the design of novel therapeutic approaches to diseases.

## Methods

### Parameters

The values of parameters in our model are determined based on previous works. The degradation rate of proteins is set $\beta$ = 0.1 /hour, equivalent to a half-life of about 7 hours [58]. The degradation of free Notch Intracellular Domain ($S$) is assumed to be $\beta_S$ = 0.5 /hour because the signal in Notch signaling decays rapidly [59]. Published data have suggested that Notch-related protein levels vary by up to a few hundred ng/ml [60], or a few thousand molecules per cell, thus the production rates of Notch ligands and receptors are assumed to be $b_L$ = 200 molecules/hour and $b_N$ = 200 molecules/hour, respectively. The trans-activation and cis-inhibition

rate were derived from previous studies [26–28], the authors estimated their parameters by referring to relevant experimental reports. We assume that the ligand oligomerization rate is $k_d = 10^{-4}/(\text{molec}^{(n-1)*}\text{hour})$, which is close to the order of magnitude of other parameters representing protein-protein interactions (Table 1).

We performed parameter sensitivity analysis by quantifying the changes of Notch activity as function of the variations of all parameters (Fig D in S1 Text). The results show that the sensitivity of most parameters is the same in three different experimental settings, thus showing a good consistency among the predictions of ligand dimerization model.

## Numerical simulations

Numerical simulation, nonlinear curve-fitting and numerical analysis were performed on MATLAB R2021a. Runge–Kutta methods were used in all scenarios.

In the simulations, the level of external ligand $L_{ext}$ is context-dependent. For the single cell exposed to fixed level of trans-ligands in Figs 6B, 6C, 6I, 8A and 8D, $L_{ext}$ = 1500 molecules. For multiple interacting cells, the level of trans-ligand is the average of ligands in neighboring cells (Fig 9). Cis-activation of Notch is always ignored when external ligands-mediated trans-activation of Notch is present.

In Fig 6I, to mimic experimental conditions [22], the initial state of Notch ligand monomer, ligand dimer, Notch receptor and Notch activity in the cell is fixed at (500, 9500, 0, 0) molecules.

In Fig 9, the peak of gradient profile of Delta production rates in wing vein patterning (Fig 9B) and the peak of ectopic production rates of ligands in dorsal-ventral patterning (Fig 9F and 9G) is $L_{max}$ = 15000 molecules/hour. For wing vein patterning (Fig 9B), the production rate of Notch is $b_N$ = 200 molecules/hour. For dorsal-ventral patterning (Fig 9C and 9D), the production rate of Notch is $b_N$ = 1000 molecules/hour, and the default production rate of ligand (Delta/Serrate) is $b_L$ = 200 molecules/hour. In Fig 9E, we varied the production rate of Notch. In Fig 9F and 9G, we included ectopic expression of Notch ligands under the basic parameter settings of Fig 9D. In Fig 9H, we simulated Notch activity mediated by ectopic ligands with different dimerization rate. The simulated cells are colored according to the level of Notch activity in each cell. Warmer color means higher Notch activity.

## Statistical significance

The statistical hypothesis testing of Notch activity in Fig 4A was performed using Student's t-test for paired two-sample.

## Experimental methods

**Cell culture, biochemistry and molecular biology.** Human embryonic kidney 293T cells and U2OS osteosarcoma cells were cultured in DMEM (Gibco) supplemented with 10% fetal bovine serum (Gibco). Cell lines were typed using short tandem repeat analysis of the DNA and all cell lines were checked for mycoplasma with the MycoAlert kit (Lonza). Transfections, lentivirus production and cell infections, Western blotting and co-immunoprecipitations have been described previously [61,62]. All lysis buffers contained a cocktail of protease inhibitors (phenylmethylsulfonyl fluoride, trypsin inhibitor, pepstatin A, leupeptin, aprotinin).

**Recombinant protein production/ *in vitro* protein:protein interaction.** Domains for recombinant protein production were cloned into the pET 28a vector in-frame to an N-terminal 6x HIS epitope. His epitope–tagged proteins were manufactured in *Escherichia coli* BL21 (DE3). Following sonication (Misonix Sonicator 3000) in 3 mls ice-cold buffer / 50 ml bacterial culture (150 mM NaCl, 2.7 mM KCl, $Na_2HPO_4$, $KH_2PO_4$, 20 mM imidazole, 10 mM $\beta$-mercaptoethanol), proteins were purified onto 50 ul of Nickel- agarose beads (Qiagen) by 3 hours

rolling at 4C. Beads were washed in 10 x 1 ml of the same buffer. Protein yields were determined by Bradford assay (Bio-Rad) and relative protein integrity and purity was determined by SDS-PAGE and Colloidal Blue staining (Invitrogen). Purified recombinant protein was incubated with 10 ul nickel beads in 1 ml of buffer for 2 hours at 4˚C with *in vitro* translated DLL4 proteins made using the TNT-coupled reticulocyte *in vitro* translation system (Promega). Beads were washed x10 with 1 ml of buffer. Proteins were separated by SDS-PAGE and associated proteins were detected by Western blot.

**Plasmid construction.** Unless otherwise stated, all cDNAs were fused in-frame with a Flag or an HA epitope tag and were cloned into the pLV lentiviral vector and pCS2 expression plasmid. Expression of these proteins was determined using antibodies directed against either epitope tag. Mutants were generated by site-directed mutagenesis using Phusion High-Fidelity DNA polymerase (Thermo fisher). All constructs were verified by Sanger sequencing (Macrogen).

**Immunofluorescence.** Immunostaining was performed as previously described [63,64] using Alexa Fluor 488 goat anti-mouse secondary antibodies (Thermo Fisher scientific). Imaging was performed with a Leica SP8 confocal microscope.

**Luciferase reporter.** Stable cell lines expressing epitope-tagged ligands and receptors were established via infection of cells with lentiviruses harbouring the appropriate ligand/receptor cDNA followed by selection with the encoded antibiotic (puromycin or neomycin). For 'cis inhibition' assays, cells co-expressing ligand, receptor and reporter were co-cultured with cells expressing ligand alone (to enable transactivation). Comparable results were obtained for co-culture ratios of 1:1, 1:2, 1:4. For transactivation analyses, cells stably expressing ligand alone were co-cultured with cells expressing receptor and reporter (1,1 ratio). For each experiment, cells were seeded in triplicate in 12-well plates. The Notch luciferase reporter harboured 10x RBPJ consensus binding sites, and was co-transfected with Renilla luciferase control plasmid. Transfection efficiencies (routinely >90%) were determined through visualization of co-transfected plasmid encoding the Tomato fluorescence reporter. Cells were lysed 36 hours post-plating, and luciferase activity was measured using a luciferase assay substrate (Promega). Luciferase activity was normalized by measuring Renilla luciferase activity (Promega). To confirm that luciferase activity was Notch-dependent, assays were performed in the presence or absence of 10 μM DAPT (Sigma). Receptor and ligand protein levels were determined by Western blotting. Experiments were performed three times.

**Antibodies and drugs.** Antibodies were obtained from the following sources: FLAG mouse M2 monoclonal (Sigma); anti-HA.11 mouse monoclonal (Covance); anti-HA rabbit polyclonal (Abcam); anti-FLAG rabbit (Sigma); anti-γ-tubulin (Sigma); anti-GFP (GeneTex); anti-His (Sigma). Anti- Calnexin (AbCam).

## Supporting information

**S1 Text. Supplementary information of this paper.** The supplementary document provides modeling details, four supplementary figures and MATLAB code for the main text. (DOCX)

**S1 Video. Multicellular simulations of Notch signalling along the wing vein in Drosophila wing.** Video corresponding to Fig 9B. (RAR)

**S2 Video. Multicellular simulations of Notch signalling at the dorsoventral boundary in Drosophila wing discs.** Video corresponding to Fig 9D. (RAR)

## Acknowledgments

We thank members of the Department of Cell & Chemical Biology of Leiden University Medical Center and the Institute of Biology Leiden of Leiden University for helpful discussions, technical advice and support, in particular Professor Peter ten Dijke, Jacob Visscher, Miguel Dickson and Yana van der Weegen.

## Author Contributions

**Conceptualization:** Daipeng Chen, Roeland M. H. Merks, David A. Baker.

**Data curation:** Daipeng Chen, Zary Forghany, Xinxin Liu, Haijiang Wang, David A. Baker.

**Formal analysis:** Daipeng Chen, Zary Forghany, Xinxin Liu, Haijiang Wang, Roeland M. H. Merks, David A. Baker.

**Funding acquisition:** Roeland M. H. Merks, David A. Baker.

**Investigation:** Zary Forghany, Xinxin Liu, Haijiang Wang.

**Methodology:** Daipeng Chen, Zary Forghany, Xinxin Liu, Haijiang Wang.

**Project administration:** David A. Baker.

**Resources:** Roeland M. H. Merks, David A. Baker.

**Software:** Daipeng Chen.

**Supervision:** Roeland M. H. Merks, David A. Baker.

**Writing – original draft:** Daipeng Chen, David A. Baker.

**Writing – review & editing:** Daipeng Chen, Zary Forghany, Xinxin Liu, Haijiang Wang, Roeland M. H. Merks, David A. Baker.

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
