## [Decision Letter · Decision Letter 0]

15 Jul 2022

Dear Dr Baker,

Thank you very much for submitting your manuscript "A new model of Notch signaling: Control of Notch receptor cis-inhibition via Notch ligand dimers" for consideration at PLOS Computational Biology.

As with all papers reviewed by the journal, your manuscript was reviewed by members of the editorial board and by several independent reviewers. In light of the reviews (below this email), we would like to invite the resubmission of a significantly-revised version that takes into account the reviewers' comments.

We cannot make any decision about publication until we have seen the revised manuscript and your response to the reviewers' comments. Your revised manuscript is also likely to be sent to reviewers for further evaluation.

Sincerely,

Martin Meier-Schellersheim

Associate Editor

PLOS Computational Biology

Jason Haugh

Deputy Editor

PLOS Computational Biology

Reviewer's Responses to Questions

**Comments to the Authors:**

Reviewer #1: The manuscript by Chen et al addresses an interesting question regarding the mechanism underlying the phenomenon of cis-inhibition of Notch receptors by Notch ligands. It has been well documented that Notch ligands, in addition to their ability to trans-activate Notch receptors in neighboring cells, can also cis-inhibit Notch receptors in the same cell. More recently, low level of cis-activation was also observed (albeit in cell culture assays only). The current manuscript suggests that while trans-activation is associated with monomer receptor-ligand binding, cis-inhibition is achieved by ligand dimerization (or oligomerization). This is an interesting hypothesis that is of significant interest to the Notch signaling and cell biology communities. The manuscript contains two parts: The first is providing biochemical evidence for Notch ligand dimerization and for its specific role in cis-inhibition but not trans-activation, and the second is a mathematical modelling part that investigates the implications of this mechanism for Notch signaling activation processes. While the manuscript is well written, and puts forward intriguing experimental and theoretical results, there are some major flaws with both the experimental aspects and the theoretical analysis. On the experimental side, there are issues with the assays for measuring cis-inhibition (detailed below). Regarding the mathematical modelling, while the formation of the equations are sound, the conclusions drawn are not supported by the analysis. In particular the claims that the model better explains experimental data compared to previous models is often substantiated by vague and descriptive statements. The differences between the old and new models are small and simply depend on parameter tuning (hence does not mean anything). I feel there is a missed opportunity here to identify differences that are substantial and that can be tested experimentally (even if not within this manuscript). At its current form the manuscript is not suitable for publication in PLOS computational Biology. However, if the authors address the experimental concerns, correct their claims, clarify the analysis and identify some significant, experimentally testable, predictions of the model, this conclusion might change.

Major concerns:

1. The Biochemical data in Fig. 1-3 show that Dll4 and Dll1 ligands can multimerize and that the interacting domains are EGF8 and MNNL. This is used in Fig. 4 to show that Dll4 lacking the EGF8 can trans-activate but not cis-inhibit. This is an important observation, however the assay used to make this observation is unclear:

a. The information provided in the text, captions, and methods is sketchy with many details missing. First, for measuring cis-inhibition one needs to provide separately ligand in-trans (plate-bound-Delta of co-cultures Delta cells) and in-cis (expression in the same cell). From the description of Fig. 4A it is not clear that this was done, as there is no trans-Delta provided. This experiments should be performed with trans-Delta. Moreover, it should be shown that this signal is Notch dependent for example by blocking gamma-secretase.

b. More information is needed on the co-culture experiments including how long prior to co-culture where the cells transfected, what was transfection method and efficiency, how were the cells moves to the co-culture dish and so on. Also here DAPT inhibition should be provided. Also a control where non-activating ligand is used (DeltaEGF1-3 or similar) should be added.

c. It is not mentioned what the error bars represents – whether the error is between repeats? Where the triplicates performed in different days? There is no statistical analysis of the results check significance?

2. It has been previously shown in Drosophila that Serrate lacking EGF repeat 4,5, or 6 can also trans-activate (in vivo) but do not cis-inhibit (Fleming 2013, PMID: 23571220). This should of course be mentioned in the manuscript. The fact that similar deletion can also remove cis-inhibition in mammalian Dll1/4 is an interesting observation that generalize the results in the manuscript.

3. Regarding the modeling. As I mentioned before the suggested equations (eq. 1-2,3) makes sense for describing the cis-inhibition mediated by ligand dimerization. However, the explanations on the different aspects of the models are severely lacking, and the conclusions drawn are either exaggerated or not substantiated. This include:

a. In Fig. 4A-C the description of what is calculated and how is unclear. It not clear which Notch activity is calculated (from which of the two cells) and whether the parameters change only in one cell or in both. A better Scheme showing the model setup of two cells would help.

b. In line 214 there are misleading statements regarding the choice of parameter. it is said that: “…the ligand dimerization rate, , based upon previously published estimates”. This is clearly not the case as Table 1 shows that this rate and other rates are assumed (just guessed). This is mentioned in other places (discussion) and it should be explained where this and other parameters come from.

c. The degradation rate of the dimer is assumed to be 10 time slower than the monomers. There is no explanation why this is assumed (no experimental data). Moreover, it should be discussed whether this difference is essential for the results presented.

d. At the end of the paragraph it is said that “ the mathematical model (1), describing ligand-oligomer-specific cis inhibition of Notch signalling, faithfully recapitulates our experimental findings”. It is really unclear which finding are referred to since in the experiments in Fig. 4 Ligand concentration and kd were not varied!

e. The only non-trivial observation in Fig. 4 is that dimer deficient ligands trans-activate but do not cis-inhibit. So, the authors should consider simulating this aspect i.e. comparing WT and mutant ligand behaviors in different models. Moreover, the models that should be compared here are the model that dimerization drive cis-inhibition (case 1) and the old model where monomers drive cis-inhibition. One could add a model where both monomers and dimers exert cis-inhibition.

f. Instead of simulating two cells, it might be better to simulate only a single cell that feels some external ligand level, and show Notch activity as a function of the cis ligand level. Then one can show this plot for different models as described in e.

g. In Figure 6 the authors compare the ability of their model to explain results obtained in ref 22. They show that their mode can qualitatively capture the experimental results obtained. However they also claim that “…it appears to more closely match the observed pattern of ligand decay”. This statement is not substantiated by any measure and should either be proven (statistically) or removed. Moreover, claiming that your model is better than the old model cannot be based on small quantitative differences that may simply reflect the choice of parameters and can be easily tuned in either model. To distinguish between models one should bring a suggested experiment that significant qualitative difference between the two models.

h. In Fig. 6C,D it is completely unclear which data is being used to perform the fit (this is again from ref 22, but cell line 1 and cell line 2 are not defined). The meaning of these fits is pointless as the original data simply show a sigmoidal dependence on external ligand without cis-inhibition (only plate bound Delta). So, there is no meaning to claiming anything regarding the new model.

i. In Fig. 7 there is a comparison of different models that include cis-activation. Here, they authors claim that their model can explain better the results from ref. 31. Once again the claim is not substantiated. The claim is: “ Notably, our ligand dimerization model generates narrower peaks using the same reference parameter values, which is closer to experimental data”. This claim has no meaning as it does not really matter what is the width of the peaks obtained in a specific realization of the simulation since it can be changed by changing the parameters. A better approach would have been to discuss the prediction of an experiment using the dimerization deficient ligand (this is what is approximately shown in Fig. 7G.

j. Fig. 8 addresses the comparison between different levels of oligomerization. It is again unclear what is plotted in Fig. 8B and how the model is compared to experiments. This is also true for Fig. 8D.

k. It is not clear why data is discussed for the first time in the discussion (models S3, S4). It should be moved to the results.

l. Along the lines in the discussion it is claimed that: “we consistently observed enhanced reporter activity under cis conditions in the presence of DLL4 ligand monomer but not DLL4 ligand dimer, which instead strongly inhibited Notch receptor activity” . This is not discussed in the results nor is it shown that this difference is statistically significance.

m. Line 316 in the discussion: “Using physiologically relevant parameters our simulations strongly favour the former case”. This is not a substantiated statement.

Minor concerns:

4. In Fig. 2 the western and commassie staining has poor resolution (hard to read the text).

5. Fig. 3B has a different font than Fig. 3A

6. Resolution of the images in Fig. 4B islow. It is not explained what is calnexin and why is it stained

7. The full code should be uploaded to a github

Reviewer #2: This paper proposes a Dll4 dimerization mechanism for cis inhibition within the Notch signaling pathway. The experiments reported here show dimerization of Dll4 in the same cell, which does not require Dll4 intracellular domain. Dll4 dimerization is required for cis-inhibition of Notch receptor activity. As a consequence, the authors modify a well-known Notch signaling model proposed by Sprinzak et al replacing dimers instead of monomers in cis inhibition while leaving the trans-activation part of the model unmodified. The modified model including dimerization is shown to agree with their experiments and some others reported in the literature.

Comments.

1. The discussion about different proposed models is very confusing. Firstly, the Sprinzak equations are frequently used and referred to in the text. They turn out to be Equations (5) of the Methods section. It would be helpful to write them before, in the Results section, and refer to them explicitly by number when needed. Otherwise, it seems one has to keep reading Sprinzak paper in parallel to this one to follow the authors’ arguments, until it is discovered that the Sprinzak equations are listed as (5) later in the manuscript. Secondly, it would be helpful if the authors make clear which of their models is best for the general case, including the contributions ‘ext’ in Eq. (4) if appropriate. One possibility would be writing a general equation instead of (2) and (4) that comprises all terms in these equations, and then specifying which terms should be set equal to zero according to which experimental information and situation.

2. The papers by Boareto et al are referred to but not really discussed. Boareto et al studied two coupled cells with equations similar to Sprinzak’s and defined sender/receiver (or tip/stalk) phenotypes. By adding Jagged dynamics to the Sprinzak equations, Boareto et al discovered new hybrid sender/receiver (tip/stalk) cellular phenotypes that turn out to play a role in sprouting and tumor angiogenesis and can be incorporated to cellular models of angiogenesis; see Vega et al, PLoS Comp. Biol. 16, e1006919, 2020. Hybrid phenotypes can be conveniently extended as tumor transition states during EMT, which have been experimentally identified by Pastushenko et al, Nature 556, 463 (2018). The present authors should comment on the consequences for hybrid sender/receiver phenotypes that possible extensions of their models to include Jagged dynamics could have.

3. Eq. (1) considers only one type of ligand oligomers of size n, and then the authors make the case that n=2. Suppose that there are concurring oligomerization processes with n=2 and higher. Would it be possible to discriminate these processes and reach the same conclusion about n=2?

The results presented in this paper are worthwhile and important for multiscale processes such as angiogenesis and tumorigenesis. The manuscript should be revised to address the previous points. Further discussion along the lines in point 2 should be added to enhance the projection of this paper.

**Have the authors made all data and (if applicable) computational code underlying the findings in their manuscript fully available?**

Reviewer #1: Yes

Reviewer #2: Yes

PLOS authors have the option to publish the peer review history of their article (what does this mean?). If published, this will include your full peer review and any attached files.

Reviewer #1: No

Reviewer #2: No
---

## [Decision Letter · Decision Letter 1]

3 Nov 2022

Dear Dr Baker,

Thank you very much for submitting your manuscript "A new model of Notch signaling: Control of Notch receptor cis-inhibition via Notch ligand dimers" for consideration at PLOS Computational Biology. As with all papers reviewed by the journal, your manuscript was reviewed by members of the editorial board and by several independent reviewers. The reviewers appreciated the attention to an important topic. Based on the reviews, we are likely to accept this manuscript for publication, providing that you modify the manuscript according to the review recommendations.

With regard to the comments of reviewer #2, please use your own discretion as to the question of how much of the mentioned work you wish to reference.

Sincerely,

Martin Meier-Schellersheim

Academic Editor

PLOS Computational Biology

Jason Haugh

Section Editor

PLOS Computational Biology

Reviewer's Responses to Questions

**Comments to the Authors:**

Reviewer #1: The revised version of the manuscript by Chen et al. is much improved compared with the original manuscript. The authors have added most of the required description on the experimental side and completely restructured the modelling part introducing new sections and comparing the predictions of the model to their results as well as to published one. The suggestion that cis inhibition requires dimerization of the ligands is quite interesting and as the authors point out have testable implications. As such this work adds a potential mechanism for how cis-inhibition works which can potentially spur significant interest in the field (maybe even some controversy). Overall, the logic and analysis of the model is sound and written in a clear manner (albeit some aspects in the last section need to be improved). I do have some comments on some aspects but trust that the authors can address these in the next version. My general feeling is that the authors could have done more on the experimental to validate several aspects of the model, but given that this manuscript is in a computational journal, it is reasonable that some predictions remain as such without the experimental tests.

The authors have pointed out that the experimental part should be short since the emphasis is on the theoretical part (this was also pointed out by reviewer 3). Given the importance of the experimental results to the theory section, I feel that it would be better if the full description of the experiments and experimental results would be in the main text, even if it means that the experimental part is expanded.

Below please find some comments on the current version that should be addressed:

1. Regarding figure 4 – I think it would add if a schematic of the co-culture experiments of Fig. 4A would be included. In the Cis(+DAPT) panel, it should be explained what Notch(-) refers to. Finally, p-values for significance should be included in the caption.

2. The Authors show in Fig.4 that ligands that lack EGF repeats does not cis-inhibit Notch2. Have they tested also the effect on Notch1? If they have, it would be a nice addition. If not, it is sufficient to mention in the discussion whether they think it would be have the same with different Notches.

3. When discussing Figure 6B it should be mentioned that the reduction in Notch activity decreases slower in model C1 (cis-inhibition through dimerization) compared to the other models. This is something that can be tested experimentally.

4. In Figure 6I the authors show that their model can reproduce the experimental result in the Sprinzak et al paper. It should be noted that also the regular cis-inhibition model matched the experiments. Namely, that the new model in this case is consistent with the experimental results as does the previous model. Hence, the new model does not explain a previously unaccounted for phenomena.

5. The steady analysis for the new model and its comparison with the old model is quite revealing. One important aspect that comes out of this analysis is that even when Notch levels are high the monomer ligand is free to activate neighboring cells. This is a key difference that its implication can be studied experimentally as the authors try to do in Figure 9. (This is just a general comment)

6. When describing the model where cis-inhibition happen in the cytoplasm, the authors should explicitly mention that only non-cis-inhibited ligands/receptors can go from the cytoplasm to the membrane. A diagram showing the assumptions of this model would be useful.

7. The conclusion of the analysis of that model in the text states that: “These simulations demonstrate that the modelling predictions do not depend on the subcellular localization of ligand dimerization and dimer mediated cis-inhibition of Notch. ”. I agree that the qualitative behavior is the same but the quantitative one seem to be different. Suggest to amend statement.

8. The new figure 9 is a nice addition to the manuscript but it lacks proper description of the assumptions taken. It should be added in the text and methods the exact assumptions and parameter values considered for each simulation.

9. Regarding the Fig. 9A-B, the authors show that their model exhibit the expected expression pattern of Notch activity. However, is there a difference between the mutual inactivation model and the new model? If not, this should clearly stated.

10. Fig. 9C-H argue that the new model can explain the pattern in the wing DV boundary while the mutual inactivation model cannot. I suspect that this depends on the assumptions taken. For example, I believe that mutual inactivation model can explain the observations if cis-inhibition strength is assumed to depend on Fringe activity (as was shown by Lebon et al eLife 2014). Moreover, the authors assume that in this case Ligand expression levels across the wing are below Notch expression levels. If it is assumed the opposite (that ligand levels are above Notch levels - I don’t think it is clear what is the case for the real system) can the observed Notch activity distinguish between the two models? This assumption and tis implication should be discussed.

11. Please note that while Ser is expressed only in the Dorsal, Delta is expresses everywhere. It is more strongly active in the ventral section where fringe is not expressed. Please amend description.

12. The patterns of overexpression of Delta and Ser predicted by the model (Fig. 9F-G) do not seem to match the experimental ones since in the experiments the two peaks of expression persist both on the ventral and the dorsal ones. See for example De Celis and Bray, Development 1997. Please explain discrepancy.

Reviewer #2: The authors of this paper claim they have revised it following recommendations from the Referees. However, this is not the case. I copy my second point below.

"2. The papers by Boareto et al are referred to but not really discussed. Boareto et al studied two coupled cells with equations similar to Sprinzak’s and defined sender/receiver (or tip/stalk) phenotypes. By adding Jagged dynamics to the Sprinzak equations, Boareto et al discovered new hybrid sender/receiver (tip/stalk) cellular phenotypes that turn out to play a role in sprouting and tumor angiogenesis and can be incorporated to cellular models of angiogenesis; see Vega et al, PLoS Comp. Biol. 16, e1006919, 2020. Hybrid phenotypes can be conveniently extended as tumor transition states during EMT, which have been experimentally identified by Pastushenko et al, Nature 556, 463 (2018). The present authors should comment on the consequences for hybrid sender/receiver phenotypes that possible extensions of their models to include Jagged dynamics could have."

What the authors have done is to suppress all references to Boareto’s papers except for citing them as the source of parameter values in their Table I. In particular, the discussion of the relevance of their results for the hybrid sender/receiver phenotypes is nowhere to be found in the revised paper. This is important because hybrid phenotypes have been found in experiments and it would be appropriate to know whether the modified model proposed by the authors is able to incorporate hybrid sender/receiver phenotypes or not. I cannot recommend publication of this version of the paper until the authors address this point in their paper, not just in their misleading response to the referees. A more professional behaviour would have been appreciated.

Reviewer #3: The authors have done a great job of addressing my comments, and their revised manuscript represents a significant improvement over the original submission. In particular, the readability of the theoretical aspects of their work has been improved by moving the statement of their extension to Sprinzak et al’s model into the results section. Also, by softening some of the statements made in the original manuscript the authors provide a more measured account of the impact of their research.

I have only two minor comments to raise (see below).

Minor Comments

• Regarding the extent to which their model is consistent with published experimental data (see lines 272-274), it would be helpful if the authors could include a statement clarifying how agreement was quantified and clarifying that they were not able to parameterise their model to experimental data. On a related note, have the authors considered performing a comprehensive parameter sensitivity analysis of their model – this might be helpful, especially with experimental design.

• It is confusing to use L and N to denote the number of Notch ligand and receptors per cell and then to use L0 and N0 to denote by production rates, as these quantities have different dimensions. It would be more natural (and less confusing) to denote by L0/tau and N0/tau the rates at which Notch ligand and receptors are produced

**Have the authors made all data and (if applicable) computational code underlying the findings in their manuscript fully available?**

Reviewer #1: Yes

Reviewer #2: None

Reviewer #3: None

PLOS authors have the option to publish the peer review history of their article (what does this mean?). If published, this will include your full peer review and any attached files.

Reviewer #1: No

Reviewer #2: No

Reviewer #3: No

Figure Files:

Data Requirements:

Reproducibility:

References:

---

## [Decision Letter · Decision Letter 2]

30 Dec 2022

Dear Dr Baker,

We are pleased to inform you that your manuscript 'A new model of Notch signalling: Control of Notch receptor cis-inhibition via Notch ligand dimers' has been provisionally accepted for publication in PLOS Computational Biology.

Best regards,

Martin Meier-Schellersheim

Academic Editor

PLOS Computational Biology

Jason Haugh

Section Editor

PLOS Computational Biology

Reviewer's Responses to Questions

**Comments to the Authors:**

Reviewer #1: The authors have addressed all comments in a satisfactory manner.

**Have the authors made all data and (if applicable) computational code underlying the findings in their manuscript fully available?**

Reviewer #1: Yes

PLOS authors have the option to publish the peer review history of their article (what does this mean?). If published, this will include your full peer review and any attached files.

Reviewer #1: No

---

## [Editor Report · Acceptance letter]

13 Jan 2023

PCOMPBIOL-D-22-00699R2 

A new model of Notch signalling: Control of Notch receptor cis-inhibition via Notch ligand dimers

Dear Dr Baker,

I am pleased to inform you that your manuscript has been formally accepted for publication in PLOS Computational Biology. Your manuscript is now with our production department and you will be notified of the publication date in due course.

With kind regards,

Anita Estes
